# Methanogenic patterns in the gut microbiome are associated with survival in a population of feral horses

Mason. R. Stothart [1,2] ✉, Philip. D. McLoughlin[3], Sarah. A. Medill[3], Ruth. J. Greuel[3], Alastair. J. Wilson [4] & Jocelyn. Poissant [1] ✉

Gut microbiomes are widely hypothesised to influence host fitness and have been experimentally shown to affect host health and phenotypes under laboratory conditions. However, the extent to which they do so in free-living animal populations and the proximate mechanisms involved remain open questions. In this study, using long-term, individual-based life history and shallow shotgun metagenomic sequencing data (2394 fecal samples from 794 individuals collected between 2013–2019), we quantify relationships between gut microbiome variation and survival in a feral population of horses under natural food limitation (Sable Island, Canada), and test metagenome-derived predictions using short-chain fatty acid data. We report detailed evidence that variation in the gut microbiome is associated with a host fitness proxy in nature and outline hypotheses of pathogenesis and methanogenesis as key causal mechanisms which may underlie such patterns in feral horses, and perhaps, wild herbivores more generally.

Gut microbiomes can influence animal health and the expression of host phenotypes[1]. Microbial residents of the intestinal tract are now recognized as integral to the adaptive programming of host physiology or capable of performing functions beyond the capacity of the host genome[2,3]. Conversely, common constituents of the gut microbiome can cause infectious disease, while disruption of the microbiome deprives animals of critical metabolic functions[4]. Growing recognition of the microbiome's integral importance underlies hypotheses that host-associated microbial communities can facilitate adaptive phenotypic change in animal populations[3,5,6]. For adaptive evolution in the host-microbiome relationship to occur, variation in the microbiome must be both causally connected to host fitness and intergenerationally transmissible. While heritability of the microbiome has been demonstrated[7]—including in the wild[8]—we still lack robust demonstrations that selection acts on naturally occurring microbiome variation within free-living animal populations.

Mounting experimental research conducted in laboratory environments provides strong evidence that host-associated microbial communities may partly underlie fitness variation within wild animal populations[9–11]. Evidence linking gut microbiome and host fitness proxies has similarly begun to emerge in wild populations, including documentation of microbiome associations with host survival in wild Seychelles warblers (*Acrocephalus sechellensis;* 148 individuals across 4 years of study)[12] and meerkats (*Suricata suricatta*; 235 individuals across 23 years of study)[13]. However, precisely estimating selection gradients in the wild requires large sample sizes and can be confounded by environmental heterogeneity within and among years. Furthermore, previous reliance on 16S rRNA gene amplicon sequencing methods has limited the characterization of non-bacterial constituents of the microbiome and made it challenging to identify the microbial mechanisms that proximately link the microbiome to host fitness. Using detailed life history data and shallow shotgun metagenomic sequencing of individual-linked fecal samples, we quantified the strength of selection acting on metagenomic variation in a free-living animal population (Sable Island feral horses; *Equus ferus caballus*). Using these results and analyses of short-chain fatty acid and dietary

[1]Faculty of Veterinary Medicine, University of Calgary, Calgary, Alberta, Canada. [2]Department of Biology, University of Oxford, Oxford, United Kingdom. [3]Department of Biology, University of Saskatchewan, Saskatoon, Saskatchewan, Canada. [4]Centre for Ecology and Conservation, University of Exeter, Penryn, United Kingdom. ✉e-mail: masonstothart@gmail.com; jocelyn.poissant@ucalgary.ca

metabarcoding data, we identified likely causal mechanisms which appear to connect the gut microbiome to a host fitness proxy (overwinter survival).

Horses were first introduced to Sable Island (Nova Scotia, Canada; Fig. 1) in the 18th century and have since remained free-living. From 2007, the ~500-horse population has been the focus of a long-term, individual-based study (Supplementary Note 1). Every year, the entirety of the population is surveyed repeatedly on foot from July–September, and all horses are identified against a comprehensive photographic database using unique color markings and morphological features. These surveys provide individual-based location, behavioral, reproductive and survival data (100% resighting probability since 2011).

Microbiome variation may not be consequential for all animals[14], but we predicted a strong relationship between the hindgut microbiome and survival in Sable Island horses. As large-bodied herbivores, horses are obligately reliant on their intestinal microbiota to digest their fibrous plant-based diet. Since Sable Island horses lack predators and competitors, host mortality is expected to be primarily attributable to pathogens and nutrient limitation in the winter. The microbiome likely mediates host responses to both these selective pressures. Nearly all mortalities in the Sable Island horse population occur during a discrete window of selection in late winter and early spring, when the abundance and quality of above-ground plant biomass is low. Survival in Sable Island horses is therefore dependent on energy reserves amassed during the preceding summer when forage is abundant. The latter coincides with our annual population surveys, during which fresh individual-linked fecal samples can be collected and immediately preserved. Notably, as hindgut fermenters, the microbiota observed in horse feces are representative of the microbial communities on which horses rely to extract nutrients from their diet[15]. Using a shallow shotgun sequencing method validated for this system[16], we jointly characterized the taxonomic and microbial gene contents of 2394 fecal samples collected from 794 horses between 2013 and 2019. Individuals in our dataset were represented by 1 random sample per year for up to 6 years (median = 3) and were at least 1 year of age at the time of sampling, since the equid microbiome stabilizes within the first year of life[17].

We tested for associations between the gut microbiome and horse survival using GLMMs (generalized linear mixed models) in which overwinter survival to the following year was modeled as a function of composite microbiome measures (alpha diversity, principal components, deviation from the average microbiome) or measures of microbe or gene family abundances in the microbiome (CLR-transformed read counts) while accounting for age, sex, sex-specific parental status, and mean longitude of horse sightings, which describes major environmental variation on Sable Island (Fig. 1). In this study, we provide strong evidence that variation in the gut microbiome is associated with host survival under free-living conditions and outline hypotheses of pathogenesis and methanogenesis as key causal mechanisms which may underlie such patterns.

## Results

### Composite measures of microbiome diversity

Highly diverse microbiomes are frequently hypothesized to be beneficial to hosts, but relationships between animal health and microbiome richness depend on the host species, environment, and microbiome type (gut, skin, reproductive, oral, reproductive, etc.)[18,19]. We observed no relationship between taxon richness and horse survival ($\beta = 0.19 \pm 0.27$ SE, $z = 0.688$, $p = 0.49$) and a negative relationship between horse survival and estimates of Shannon evenness ($\beta = -0.16 \pm 0.07$ SE, $z = -2.391$, $p = 0.02$). We similarly observed negative relationships between horse survival and microbial gene family richness ($\beta = -0.45 \pm 0.15$ SE, $z = -3.058$, $p = 2.2e^{-03}$) and Shannon evenness ($\beta = -0.16 \pm 0.06$ SE, $z = -2.506$, $p = 0.01$). Microbial richness in the gut microbiome may thus be less important for host fitness than underlying patterns of community composition and metabolic function in this population[18]. This interpretation is supported by significant associations between horse survival and Aitchison distance principal components of microbiome community and gene family composition (Fig. 2; Supplementary Table 1, 2).

Because host-microbiome relationships may tend to exist near an adaptive peak, it has been hypothesized that deviation of microbiomes from the population average could signal dysbiosis[4]. This hypothesis was partly supported by a negative association between horse survival and the Aitchison distance separating an individual's microbiome from the population's within-year average community composition ($\beta = -0.15 \pm 0.06$ SE, $z = -2.537$, $p = 0.01$); however, a comparable relationship was not observed with respect to gene family composition ($\beta = -0.002 \pm 0.06$ SE, $z = -0.03$, $p = 0.98$). These results indicate that horse mortality may be linked to specific microbial gene content in the microbiome, but that mortality-associated patterns in gene family contents may derive from different microbiota or alternative community states.

To evaluate whether microbiome data were informative of horse survival to the following year, while simultaneously evaluating the comparative explanatory power of taxonomic versus gene family measures, we used AIC model selection to compare: (1) a base model containing only non-microbiome terms; (2) base models plus principal components from either microbial community or gene family

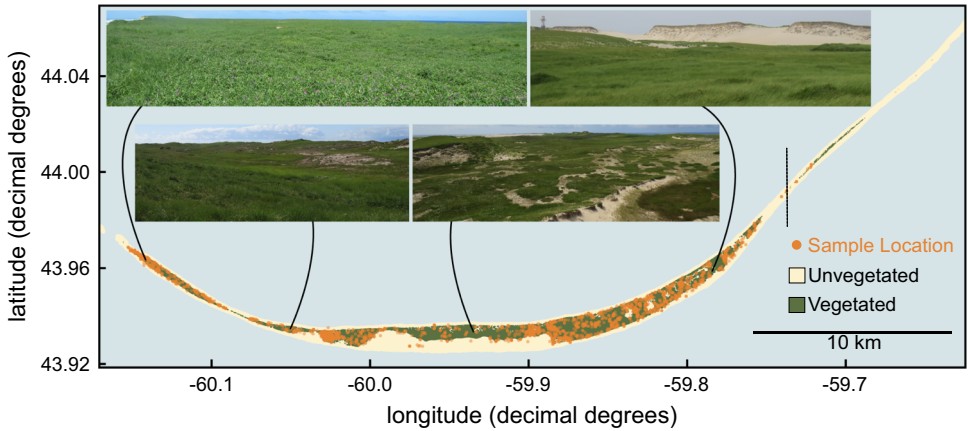

**Fig. 1 | A map of Sable Island (Nova Scotia, Canada) with representative photographs of the landscape.** Sample collection locations marked with orange points and a dashed vertical line indicated the easternmost portion of Sable Island which has been submerged since 2017. Photos © Mason R. Stothart. Source data are provided as a Source Data file.

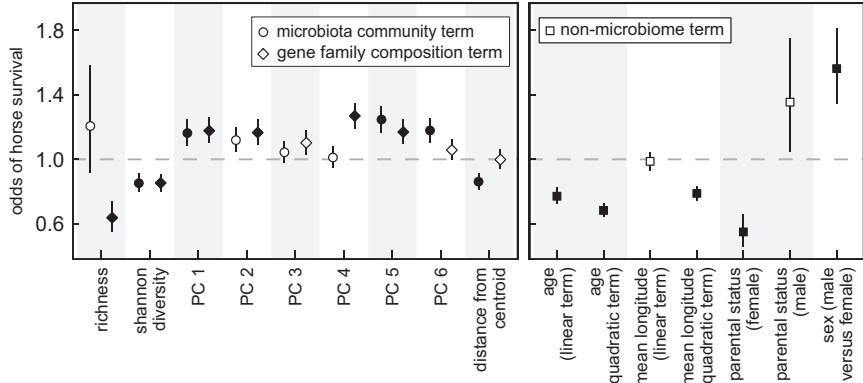

**Fig. 2 | Relationships between Sable Island horse survival and composite fecal microbiome, life history, or environmental terms.** Points denote effect estimates from generalized linear mixed models (± standard error bars) and describe relationships between the odds of overwinter survival and categorical variables, or the expected effect of a one standard deviation increase in the magnitude of continuous variables. Principal components (PCs) obtained from Aitchison distance ordinations. Distance from centroid denotes the Aitchison distance separating samples from the within-year centroid. Closed points denote statistically significant effects among models which included 2394 samples from 794 individuals. Source data are provided as a Source Data file.

ordinations which significantly predicted horse survival; and (3) base models plus microbiome distances from the population's within-year centroid from ordinations of either microbial community or gene family composition. The model including gene family principal components was the most parsimonious (AIC$_w$ = 0.88; Supplementary Table 3), followed by the model which included principal components from community ordinations (△AIC = 3.89, AIC$_w$ = 0.12). Distance-from-centroid models (community: △AIC = 18.18; gene family: △AIC = 24.58) and the base non-microbiome model (△AIC = 22.58) performed comparatively poorly. Collectively, these results suggest that microbiome data can provide an informative bioindicator of animal health in the wild and that measures of microbial gene contents may be marginally more informative than measures of community composition.

## Metagenomic predictors of horse survival

Associations between composite microbiome measures and horse survival support a widely hypothesised link between host fitness and the microbiome in the wild but provide limited insight as to which microbes in the horse gut (Fig. 3a) underlie these associations, or the proximate mechanisms involved. By individually testing relationships between horse survival and microbial taxa and gene families, we identified 155 of 1574 taxa (51 positive, 104 negative associations, Supplementary Data 1) and 27 of 1416 gene families (23 positive, 4 negative associations, Supplementary Data 2) significantly associated with horse survival, after false discovery rate correction (FDR, Benjamini and Hochberg)[20].

These results provided evidence for at least 2 major mechanisms which we hypothesize underlie microbiome-survival associations in this system. First, survival of Sable Island horses was negatively associated with a collection of putative opportunistic pathogens (e.g., *Clostridium* spp., *Erysipelothrix rhusiopathiae*, *Eggerthella lenta*, *Hathewaya proteolytica*, *Streptococcus equi*) and the only gene families that were negatively associated with horse survival—sortase B, acid phosphatase, and nitrite reductase genes—are all recognized as virulence factors in bacterial infection[21–23]. Second, we observed an array of taxonomic and functional patterns that suggest horse survival may be associated with methanogenesis in the hindgut. Specifically, taxa associated with methane production generally decreased the probability of horse survival, while microbiota and gene families involved in methanotrophy or the competitive inhibition of methanogenesis had the opposite effect. Methanogenesis can prevent H$_2$ from accumulating in the gut, where at high partial pressures it inhibits critical metabolic reactions; but it does so at a significant energetic cost. For example, among herbivorous mammals, methane emissions account for a loss of 2%–15% of gross energy intake[24–26]. We posit that comparable methane-related energy deficits would be consequential in Sable Island horses for which a depletion of energy reserves in the winter is the most common cause of mortality[27].

A relationship between methanogenesis and host performance was most apparent from the strong, negative associations between horse survival and taxa known to produce methane or promote its synthesis, including archaea of the methanogenic genus *Methanobrevibacter*[28] and bacteria associated with high methane emissions in domestic animals (e.g., *Mogibacterium*, *Pyramidobacter*, *Christensenella*, *Anaerostipes*, *Ruminococcus*, *Aminipila*; Supplementary Data 1; Fig. 3b)[28–32]. As hydrogenotrophic methanogens, *Methanobrevibacter* produce methane from CO$_2$ and H$_2$, while the aforementioned bacteria can increase methanogenesis by providing *Methanobrevibacter* with rate limiting H$_2$—a byproduct of plant fiber degradation and common fermentation pathways that end in the short-chain fatty acids (SCFAs) acetate and butyrate[28,33]. Furthermore, some acetate and butyrate producing bacteria (*e.g.*, mortality-associated *Christensenella minuta*; β = −0.18 ± 0.06 SE, $z$ = −2.99, $q$ = 0.04) directly support methanogenesis by actively cross-feeding *Methanobrevibacter* with rate-limiting H$_2$[31]. Partly owing to methanogenic energy losses, the methanogenic syntrophy between *Christensenella* and *Methanobrevibacter* has been identified as among the strongest microbiome-based predictors of a lean body mass index in humans[31]. A similar microbiome-derived metabolic phenotype is likely to be maladaptive in nutritionally stressed wildlife like Sable Island horses.

Metagenomic patterns linked to the competitive interference of methanogenesis in the microbiome were positively associated with horse survival. This includes positive associations between horse survival and bacteria capable of sulfate or nitrate reduction (e.g., *Actinoplanes*, Desulfobacteraceae, Nitrospirae, *Paracoccus*, *Sulfuricella*; Supplementary Data 1; Fig. 3b)[34–38], as well as the normalized abundance of reads mapped to hydrogen dehydrogenase genes (β = 0.26 ± 0.06 SE, $z$ = 4.066, $q$ = 0.03), a key enzyme used by bacteria to reduce sulfate in a reaction that consumes H$_2$[39]. Sulfate and nitrate reduction operate as non-methane hydrogen sinks[28], and therefore, nitrate and sulfate reducing bacteria can inhibit methanogenesis by competing with methanogens for H$_2$.

The ability for sulfate and nitrate reduction to reduce methanogenic energy loss may also partly explain why some of the strongest positive associations with survival we observed were in response to genera of Bacteroidota (formerly Bacteroidetes), which contains many intestinal mucus foragers, including *Bacteroides*, *Muribaculum*, and *Alistipes* (Fig. 3b)[40]. In the mouse gut, *Bacteroides* can cross-feed sulfate reducing bacteria with rate-limiting sulfate salvaged from mucin glycans[38], and in cultures of human fecal bacteria, the addition and

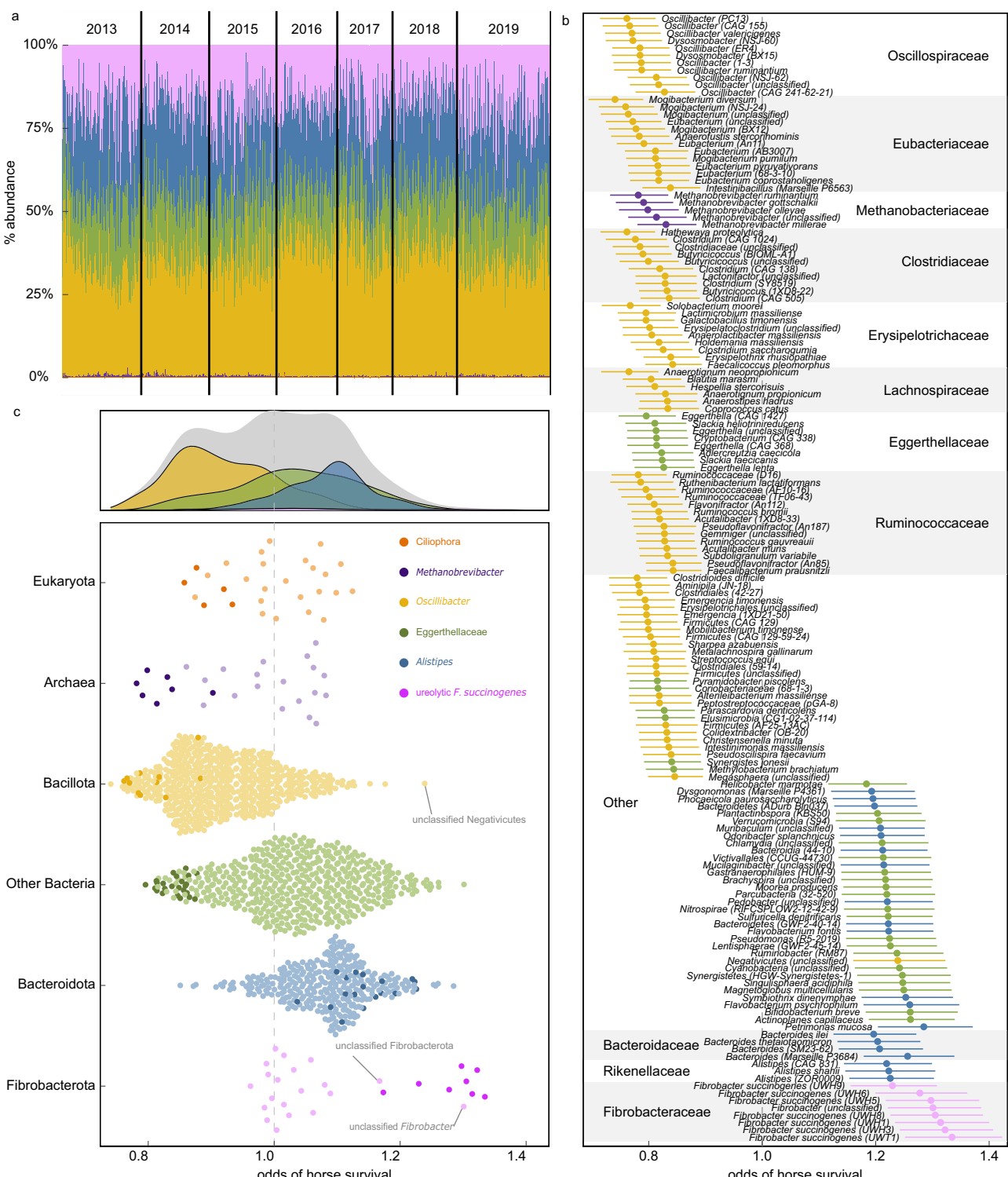

**Fig. 3 | Fecal microbiome associations with Sable Island horse survival.**
**a** Stacked bar plot in ascending order of collection date and colored by coarse taxonomic groupings described in panel 'c', **b** Generalized linear mixed model estimated effects of a one standard deviation increase in the centered log ratio (CLR) transformed abundance of microbiota on the odds of horse overwinter survival, colored by taxonomic groupings presented in panel 'c'. Points denote microbiota-specific estimates ± standard error bars estimated from analysis of 2394 samples from 794 individuals. Only significant effects after FDR adjustment displayed, **c** Generalized linear mixed model estimated effects of a one standard deviation increase in the CLR-transformed abundance of microbiota on the odds of horse overwinter survival, with points representing microbiota-specific estimates. Source data are provided as a Source Data file.

subsequent degradation of mucin glycans by Bacteroidota is necessary for facilitating the competitive inhibition of methanogenesis by sulfate reducing bacteria[37].

The general trend whereby Bacteroidota were positively associated with horse survival may also be partly attributable to the prevalence of metabolic pathways for propionate formation among this phylum[41,42]. Unlike hexose fermentation to acetate or butyrate, propionate formation results in a net sequestration of hydrogen and can therefore competitively reduce methanogenesis[28,32,33]. This might explain the divergent associations with survival between two of the

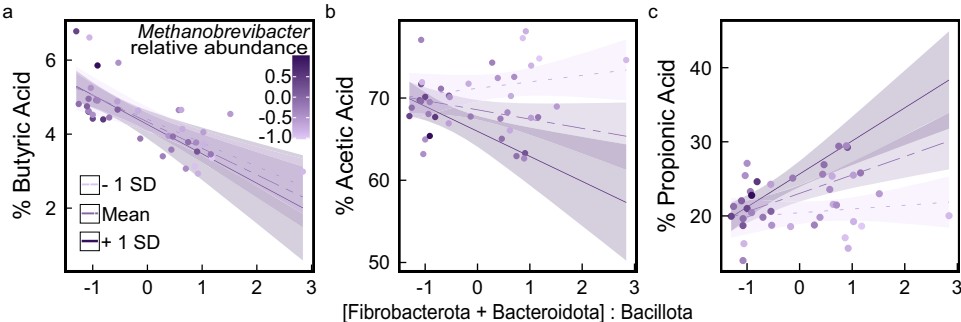

**Fig. 4 | Percent short-chain fatty acid contents relative to an interaction between *Methanobrevibacter* and the ratio of Fibrobacterota and Bacteroidota to Bacillota relative abundance.** [Fibrobacterota + Bacteroidota]:Bacillota versus (**a**) % butyric acid, (**b**) % acetic acid, and (**c**) % propionic acid, including interactions with *Methanobrevibacter* relative abundance (log-transformed, centered and variance standardized). Model predicted lines denote the effect that the mean (purple irregular dashed line), one standard deviation (SD) decrease (light purple dotted line), or one SD increase (dark purple solid line) in *Methanobrevibacter* log-transformed relative abundance has on the relationship between [Fibrobacterota + Bacteroidota]:Bacillota and % SCFA with 95% confidence interval shading. Source data are provided as a Source Data file.

most abundant bacterial phyla in the horse hindgut (Bacteroidota and Bacillota; Fig. 3a), since Bacillota (formerly Firmicutes) rarely produce propionate as a major SCFA[42]. Propionate producing Negativicutes are a well-known exception to this general pattern[41,42], but were also notable in our analyses as the only Bacillota which had significant positive associations with horse survival ($\beta = 0.21 \pm 0.07$ SE, $z = 3.266$, $q = 0.02$; Fig. 3b).

The strongest positive associations with overwinter survival were observed in relation to *Fibrobacter succinogenes* (Fig. 3c), an abundant plant-fiber degrading bacteria which produce acetate and succinate as its primary fermentation end-products[43]. Like sulfate and nitrate reducers, succinate producers compete with methanogens for electron donors ($H_2$ and formate), while reducing oxaloacetate to malate and fumarate to succinate, which is rapidly converted to propionate in the gut (predominately by Bacteroidota and Negativicutes). *F. succinogenes* demonstrate the ability to competitively interfere with methanogenesis in vitro[44] and in vivo[45–47], and in doing so, redirect energy away from methane emissions and towards SCFA synthesis. Competitive interference with methanogenesis may therefore contribute to the association we observed between *F. succinogenes* and horse survival. This interpretation is supported by a positive association between horse survival and the abundance of reads which mapped to phosphoenolpyruvate carboxykinase genes ($\beta = 0.24 \pm 0.07$ SE, $z = 3.443$, $q = 0.04$), since this enzyme catalyzes the interconversion of phosphoenolpyruvate to oxaloacetate in the first step towards succinate synthesis. We predict that a greater abundance of phosphoenolpyruvate carboxykinase gene content signals more succinate (and therefore propionate) versus acetate and butyrate formation in the horse gut.

## SCFA tests of metagenomic predictions

To test predictions derived from our shotgun metagenomic dataset related to changes in acetate, butyrate, and propionate concentrations, we generated a short-chain fatty acid dataset from a subset of 40 fecal samples. All samples were collected from adult males in 2019, spanned extremes in Fibrobacterota (formerly Fibrobacteres) relative abundance and included samples from both horses that lived or died during the following winter.

First, we sought to test whether the abundance of Fibrobacterota and Bacteroidota (primary succinate and propionate producers in the gut, respectively) relative to Bacillota (primary butyrate producers in the gut) was associated with shifts in the balance of SCFAs, as we predicted from metagenomic patterns. The ratio of Fibrobacterota and Bacteroidota versus Bacillota ([Fibrobacterota + Bacteroidota]: Bacillota) was strongly negatively associated with % butyric acid of

total SCFA content as predicted ($\beta = -0.66\% \pm 0.10\%$ SE, $t = -6.462$, $p < 0.001$; Fig. 4a), but contrary to our predictions, not % acetic acid ($p = 0.34$; Fig. 4b) or % propionic acid ($p = 0.26$; Fig. 4c). This could be because low partial pressures of $H_2$ are thermodynamically favorable for acetate formation but rate-limiting for propionate synthesis which require hydrogen incorporating steps[31,48]. Low $H_2$ availability can likewise slow rates of methanogenesis[31,48], perhaps explaining the negative association between log-transformed *Methanobrevibacter* relative abundance and % acetic acid ($\beta = -1.75\% \pm 0.58\%$ SE, $t = -3.00$, $p = 0.005$), but a positive association with % butyric acid ($\beta = 0.31\% \pm 0.14\%$ SE, $t = -2.215$, $p = 0.03$) and a non-significant positive trend with % propionic acid ($\beta = 1.04\% \pm 0.60\%$ SE, $t = 1.721$, $p = 0.09$).

Metabolic transitions by Fibrobacterota and Bacteroidota from acetate to succinate or propionate formation, as well as *Methanobrevibacter* growth, may both depend on $H_2$ availability in the gut[48]. We found no evidence that % butyric acid was affected by an interaction between log-transformed *Methanobrevibacter* relative abundance and [Fibrobacterota + Bacteroidota]:Bacillota ratio ($p = 0.50$; Fig. 4d). However, interactions were observed whereby, at higher *Methanobrevibacter* abundance, [Fibrobacterota + Bacteroidota]:Bacillota was more strongly negatively and positively associated with % acetic acid ($\beta = -1.9\% \pm 0.57\%$ SE, $t = -3.374$, $p = 0.002$; Fig. 4e) and % propionic acid ($\beta = 1.98\% \pm 0.53\%$ SE, $t = 3.748$, $p < 0.001$; Fig. 4f), respectively. After accounting for these interactions, % propionate ($\beta = 2.48\% \pm 0.60\%$ SE, $t = 4.134$, $p < 0.001$), but not % acetate ($p = 0.09$), was associated with [Fibrobacterota + Bacteroidota]:Bacillota.

The ratio of [Fibrobacterota + Bacteroidota]:Bacillota was not associated with total SCFA concentration in feces ($p = 0.45$), but a negative relationship was observed in response to *Methanobrevibacter* log-transformed relative abundance ($\beta = -6.053 \pm 2.872$ mmol/kg SE, $t = -2.108$, $p = 0.04$). As above, an interaction existed whereby higher values of [Fibrobacterota + Bacteroidota]:Bacillota weakened the negative association between *Methanobrevibacter* and total SCFA content ($\beta = 4.062 \pm 1.66$ mmol/kg SE, $t = 2.441$, $p = 0.02$), possibly as a result of shifting SCFA production from acetate to hydrogen-sequestering succinate and propionate formation, as suggested by our analysis of % SCFA, and predicted by kinetic models of SCFA formation[48]. These results support our interpretation that propionate production from succinate may act as a hydrogen sink capable of mitigating methanogenic energy loss among Sable Island horses.

As predicted, similar patterns in SCFA profiles were observed in response to survival-associated phosphoenolpyruvate carboxykinase gene hit abundance (Supplementary Fig. 1). However, phosphoenolpyruvate carboxykinase gene abundance was not associated with total SCFA concentration ($p = 0.35$), and a significant interaction

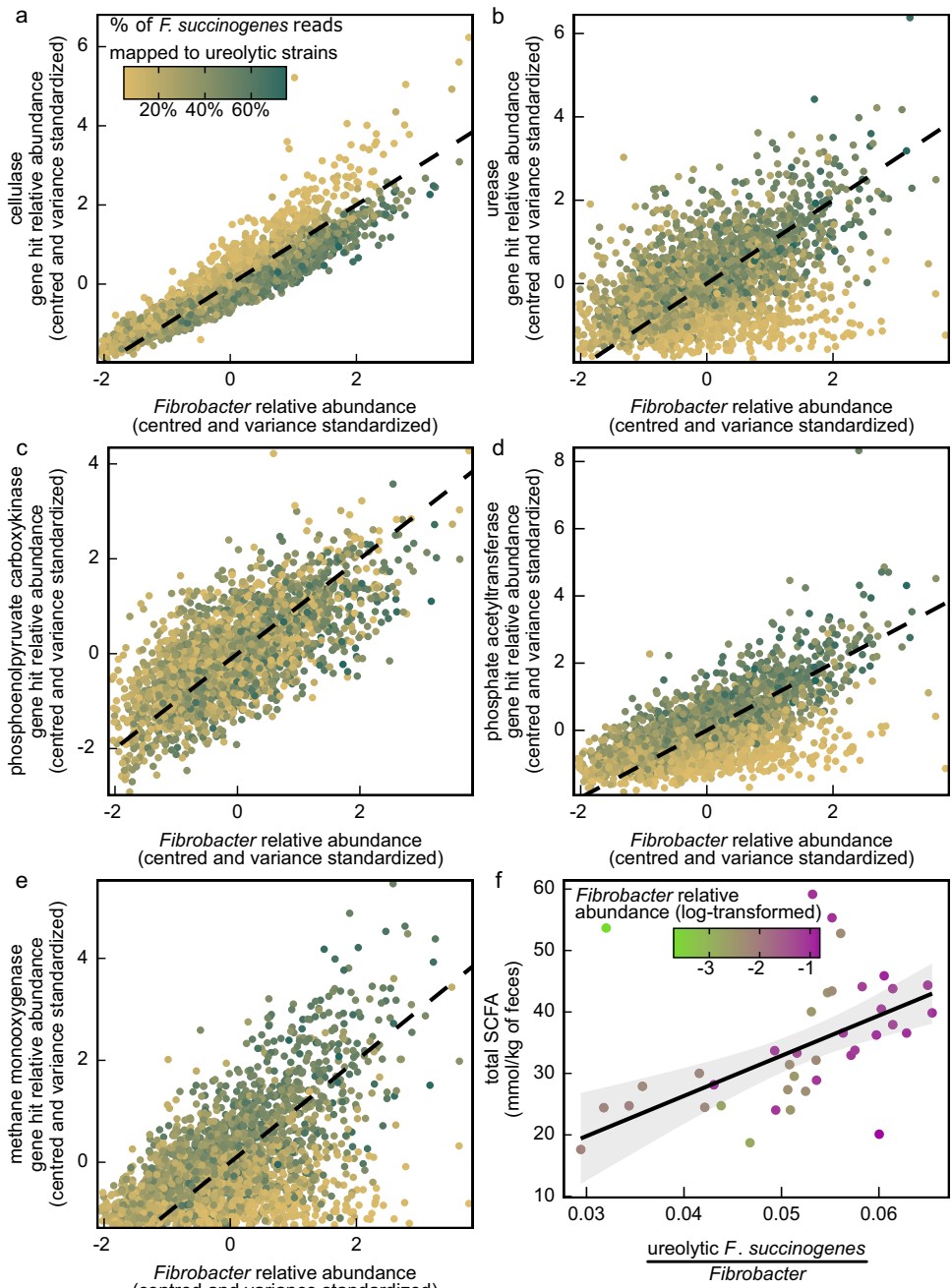

**Fig. 5 | Relationships between ureolytic *Fibrobacter succinogenes* strains, select microbial gene families, and short-chain fatty acid (SCFA) content within the Sable Island horse fecal microbiome.** *Fibrobacter* relative abundance versus (**a**) cellulase, (**b**) urease, (**c**) phosphoenolpyruvate carboxykinase, (**d**) phosphate acetyltransferase, and **e** methane monooxygenase gene hit relative abundance (centered and variance standardized) with dashed 1:1 line. Points colored by the % of reads mapped to *F. succinogenes* that belonged to ureolytic strains. **f** Proportion of *Fibrobacter* reads mapped to ureolytic strains of *F. succinogenes* versus total SCFA concentration within fecal samples. Solid line denotes the line of best fit with 95% confidence interval shading, omitting a statistically significant outlier with abnormally low *Fibrobacter* relative abundance (<2%; outlier point shown). Points colored by log-transformed *Fibrobacter* relative abundance. Source data are provided as a Source Data file.

with *Methanobrevibacter* was not observed (p = 0.09). Therefore, phosphoenolpyruvate carboxykinase alone does not account for the maintenance of SCFA production under conditions of high *Methanobrevibacter* abundance.

### Evidence for strain-specific effects

All *F. succinogenes* degrade plant fibers and produce succinate[43], but only reads which mapped to strains belonging to a monophyletic clade of *F. succinogenes* were significantly associated with survival (Fig. 3b,c).

Strains within this 'clade C' group of *F. succinogenes* have a reduced capacity to degrade plant fibers when compared to conspecifics but possess urease genes and can grow on urea as a substrate[43]. Relationships between *F. succinogenes* strain abundances and cellulase and urease gene hit abundances in our dataset are aligned with these previously described niche differences between strains (Fig. 5a, b)[43]. Like cellulase, but unlike urease, phosphoenolpyruvate carboxykinase gene hits were positively associated with both ureolytic and non-ureolytic *F. succinogenes* strain abundance (Fig. 5c), suggesting that

both ureolytic and non-ureolytic *F. succinogenes* use this enzyme to produce succinate. Therefore, differences in phosphoenolpyruvate carboxykinase abundance may not be responsible for strain-specific associations with horse survival. Ureolytic gut microbiota improve host nitrogen retention and facilitate hibernation in mammals[49]. We therefore hypothesized that strain-specific associations of *F. succinogenes* with horse survival might be attributed to the combined abilities of ureolytic *F. succinogenes* to simultaneously degrade plant fibers and recycle nitrogen; however, we observed little support for this hypothesis, since neither cellulase ($\beta = 0.19 \pm 0.07$ SE, $z = 2.849$, $q = 0.11$) nor urease ($\beta = 0.20 \pm 0.06$ SE, $z = 3.057$, $q = 0.09$) gene hits were significantly associated with horse survival.

Horse survival was instead strongly positively associated with phosphate acetyltransferase gene abundance ($\beta = 0.27 \pm 0.07$ SE, $z = 4.031$, $q = 0.02$), and phosphate acetyltransferase gene hits were positively associated with ureolytic (but not non-ureolytic) *F. succinogenes* strain abundance (Fig. 5d). Phosphate acetyltransferases catalyze the formation of acetylphosphate, which can be used as a substrate for acetate formation, or for the post-translational modification of proteins via lysine acetylation. Accordingly, phosphate acetyltransferase abundance is strongly positively correlated with the abundance of lysine acetylated proteins in the microbiome[50]. Phosphoenolpyruvate carboxykinase is the most abundant protein in the mammalian microbiome to undergo lysine acetylation[50], and the acetylation of phosphoenolpyruvate carboxykinase causes it to strongly favor the oxaloacetate (and therefore succinate) forming direction of the reversible reaction that this enzyme catalyzes[51]. Accordingly, phosphoenolpyruvate carboxykinase acetylation is hypothesized to shift the balance of SCFA synthesis in the gut microbiome away from acetate and butyrate, and towards succinate and propionate[50]. We hypothesize that phosphoenolpyruvate carboxykinase may more frequently be acetylated in ureolytic strains of *F. succinogenes*. If so, they may favor succinate over acetate as a fermentation end-product, placing them in more direct competition with methanogens for $H_2$, when compared to non-ureolytic strains.

Interestingly, methane monooxygenase genes had the strongest positive association with horse survival ($\beta = 0.30 \pm 0.06$ SE, $z = 4.772$, $q = 0.001$) of all identifiable gene families. This enzyme is used by methanotrophs to catalyze the oxidation of methane to methanol[52]. Like urease and phosphate acetyltransferase, methane monooxygenase gene hits were positively associated with the abundance of survival-associated ureolytic (but not non-ureolytic) *F. succinogenes* strains (Fig. 5e). Therefore, in addition to perhaps competitively inhibiting methanogenesis, ureolytic *F. succinogenes* might be capable of directly recapturing energy otherwise lost to methanogenesis. Laboratory experimentation and in-depth 'omics analyses will be required to determine if ureolytic *F. succinogenes* can occupy a facultatively methanotrophic niche or facilitate methanotrophy in the horse hindgut.

Like [Fibrobacterota + Bacteroidota]:Bacillota, total SCFA concentration was shaped by an interaction between *Methanobrevibacter* and ureolytic *F. succinogenes* abundance ($\beta = 3.78 \pm 1.80$ mmol/kg SE, $t = 2.102$, $p = 0.04$), but a comparable interaction was not observed with respect to non-ureolytic *F. succinogenes* strains ($p = 0.92$). The proportion of total *Fibrobacter* reads belonging to ureolytic *F. succinogenes* strains were also strongly positively associated with absolute concentrations of total SCFAs ($\beta = 5.86 \pm 1.35$ mmol/kg SE, $t = 4.350$, $p = 0.001$), propionic acid ($\beta = 1.57 \pm 0.38$ mmol/kg SE, $t = 4.138$, $p < 0.001$), and acetic acid ($\beta = 3.96 \pm 0.96$ mmol/kg SE, $t = 4.146$, $p < 0.001$), but not butyric acid ($p = 0.08$). These results support the hypothesis that survival-associated *F. succinogenes* may more efficiently competitively interfere with methanogenesis within the horse gut than non-ureolytic strains.

A significant interaction was also observed, whereby phosphate acetyltransferase gene abundance modulated the relationship

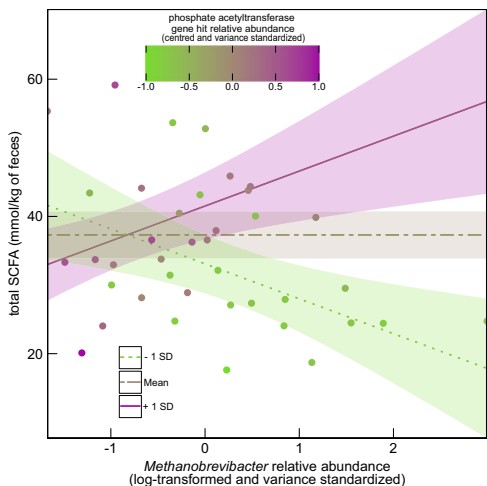

**Fig. 6 | Total short-chain fatty acid (SCFA) concentration in the Sable Island horse fecal microbiome modeled as a response to an interaction between *Methanobrevibacter* and phosphate acetyltransferase gene hit abundance.** *Methanobrevibacter* relative abundance estimates were log-transformed, centered, and variance standardized. Model predicted lines denote the effect that the mean (brown irregular dashed line) and a one standard deviation (SD) decrease (green dotted line) or increase (magenta solid line) in phosphate acetyltransferase gene relative abundance has on the relationship between *Methanobrevibacter* relative abundance and total SCFA content, with 95% confidence interval shading. Source data are provided as a Source Data file.

between *Methanobrevibacter* and total SCFA concentration ($\beta = 4.96 \pm 1.88$ mmol/kg SE, $t = 2.624$, $p = 0.01$; Fig. 6). This interactive effect was observed with respect to concentrations of acetic acid ($\beta = 2.635 \pm 1.252$ mmol/kg SE, $t = 2.104$, $p = 0.04$) and propionic acid ($\beta = 1.80 \pm 0.47$ mmol/kg SE, $t = 3.821$, $p < 0.001$) but not butyric acid ($p = 0.1$). The relative abundance of phosphate acetyltransferase gene hits was also positively associated overall with total SCFA ($\beta = 4.18 \pm 1.81$ mmol/kg SE, $t = 2.094$, $p = 0.04$) and propionic acid ($\beta = 1.88 \pm 0.45$ mmol/kg SE, $t = 3.801$, $p < 0.001$) concentrations, but not acetic ($p = 0.13$) or butyric ($p = 0.76$) acid. These results support predictions from our metagenomic dataset that phosphate acetyltransferases may partly mediate *F. succinogenes* competition with *Methanobrevibacter* for $H_2$, by facilitating succinate, and indirectly propionate, production.

## Change in the microbiome preceding host death

While microbiome variation predicts the overwinter survival of horses, it is unclear from our preceding analyses whether mortality-associated features are repeatable within individuals, change only in the year immediately preceding death, or emerge over longer timescales; for example, if immunosenescence contributes to dysregulation in the microbiome[53]. In instances where the microbiome changes prior to mortality events, it is further unclear whether beneficial and deleterious features shift together or asynchronously. Additionally, our initial analytical approach would not have detected survival-associated features that reach their maximum or minimum multiple years prior to host death. To address these gaps, we (i) used restricted maximum likelihood models to estimate the repeatability of CLR-transformed abundances of microbes or gene families within the microbiome, and (ii) compared competing models which used either overwinter survival (yes/no; abrupt change model) or the number of years before death (multi-year change model; maximum 5 years prior to death) as fixed effects for explaining variation in microbiome feature abundance.

Horse identity accounted for $10\% \pm 6.0\%$ SD of variance in microbe abundance (max = 38%) and $5\% \pm 4.5\%$ SD of variance in gene family abundance (max = 28%) on average. However, the relationships

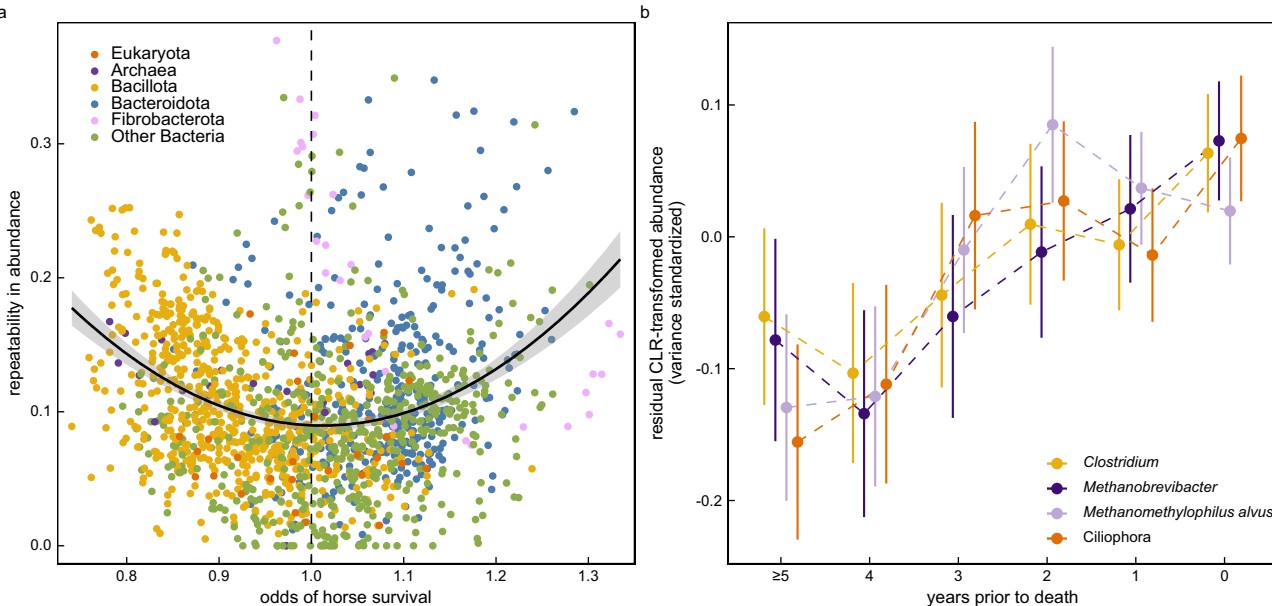

**Fig. 7 | Patterns of repeatability and multi-year change in the Sable Island horse fecal microbiome in the years preceding horse death. a** Within-individual repeatability of microbe centered log ratio (CLR) transformed abundance estimates relative to the estimated effect of a one standard deviation increase in the CLR-transformed abundance of microbiota on the odds of overwinter survival. Points denote taxa-specific model estimates and the solid black line shows the best fit quadratic relationship with 95% confidence interval shading. **b** Residual *Clostridium*, *Methanobrevibacter*, *Methanomethylophilus alvus*, and Ciliophora centered and variance standardized CLR-transformed abundance in the years preceding horse death. Points denote the mean value ± standard error bars. Residual abundance estimates derived from general linear mixed models containing 1127 fecal samples from 418 individuals that died over winter between 2013 and 2022, and terms for age (2nd order polynomial), longitude of sample collection (2nd order polynomial), date, sex, and a sex × parental status interaction as fixed effects, as well as year and horse identity as random effects. Source data are provided as a Source Data file.

between microbiome features and horse survival were not independent of repeatability estimates. The microbes most strongly associated with horse survival tended to be more repeatable within individuals across years (linear term: $p = 0.10$; quadratic term: $\beta = 0.68 \pm 0.06$ SE, $t = 11.954$, $p < 0.001$; Fig. 7a), as were gene families (Supplementary Fig. 2). Although microbiomes can be dynamic, these results suggest that the features most strongly connected to horse survival may be relatively stable.

Overall, we observed stronger evidence for abrupt rather than multi-year changes preceding death for most taxonomic (mean △AIC = $-2.38 \pm 1.51$ SD, median △AIC = $-2.47$; Supplementary Data 3; Supplementary Fig. 3) and gene family (mean △AIC = $-2.89 \pm 1.89$ SD, median △AIC = $-2.75$; Supplementary Data 4) features. This could indicate that when change occurs in survival-associated microbiome features, it tends to do so only in the year immediately preceding death, although selective disappearance may also partly underlie these patterns (i.e., individuals with maladaptive microbiomes may be less likely to be sampled multiple years prior to death). Nonetheless, outliers from these trends suggested that some features in the hindgut linked to methanogenesis may instead increase across lengthier multi-year timescales.

Of the 25 gene families significantly associated with years before death (after FDR adjustment) and which showed more support for multi-year than abrupt change (△AIC > 0; Supplementary Data 4), 24 increased in abundance preceding death. One of the largest multi-year increases preceding death was with respect to pyruvate synthase gene hit abundance ($\beta = 0.06 \pm 0.02$ SE, $t = 3.22$, $q = 0.03$). Pyruvate synthase catalyzes the interconversion of pyruvate to acetyl-CoA in the first step of the most common pathways of acetate and butyrate synthesis[54], and in doing so, releases $H_2$. Much as phosphoenolpyruvate carboxykinase gene abundance could signal succinate synthesis in the microbiome, increases in pyruvate synthase might indicate a shift away from succinate and propionate (net hydrogen sinks), towards acetate and butyrate (net hydrogen sources)[28]. We find support for

this interpretation among our SCFA data, in which pyruvate synthase gene abundance is positively associated with ratios of butyric:propionic ($\beta = 0.031 \pm 0.009$ SE, $t = 3.518$, $p = 0.001$) but not acetic:propionic (p = 0.54).

Significant increases in Vinylacetyl-CoA Delta-isomerase ($\beta = 0.06 \pm 0.02$ SE, $t = 3.024$, $q = 0.03$) and 4-hydroxybutanoyl-CoA dehydratase ($\beta = 0.05 \pm 0.02$ SE, $t = 2.87$, $q = 0.04$) gene hit abundance in the years preceding host death provided further evidence that Sable Island horse mortality could be associated with a subversion of succinate (and therefore propionate) as a non-methane hydrogen sink. Vinylacetyl-CoA Delta-isomerase and 4-hydroxybutanoyl-CoA dehydratase are key participants in the fermentation of succinate to butyrate. In support of this interpretation, we find that butyric:propionic acid is positively associated with both Vinylacetyl-CoA Delta-isomerase ($\beta = 0.027 \pm 0.009$ SE, $t = 2.953$, $p = 0.005$) and 4-hydroxybutanoyl-CoA dehydratase ($\beta = 0.031 \pm 0.009$ SE, $t = 3.597$, $p < 0.001$) abundance. The fermentation of succinate to butyrate is relatively uncommon and is perhaps best characterized among Bacillota belonging to the genus *Clostridium*[55]. A total of 69 of 1574 taxa showed greater evidence for multi-year rather than abrupt change (△AIC > 0; 39 taxa increased and 30 taxa decreased), including many *Clostridium* and *Clostridium*-associated co-abundant gene groups (Supplementary Fig. 3) which collectively increased in abundance in the years preceding horse death ($\beta = 0.04 \pm 0.02$ SE, $t = 2.408$, $p = 0.02$; Fig. 7b).

Like numerous *Clostridium*, the majority of *Methanobrevibacter* displayed slightly stronger evidence for multi-year rather than abrupt increases in abundance preceding death (median △AIC = 0.66; Supplementary Fig. 3). Increases in abundance were statistically significant among 5 *Methanobrevibacter* (Supplementary Data 3) and *Methanobrevibacter* overall ($\beta = 0.05 \pm 0.02$ SE, $t = 2.921$, $p = 0.004$; Fig. 7b). However, the strongest evidence for multi-year rather than abrupt change preceding death was observed with respect to *Candidatus Methanomethylophilus alvus* (△AIC = 5.36; $\beta = 0.05 \pm 0.02$ SE, $t = 2.730$, $q = 0.02$; Fig. 7b). Unlike hydrogenotrophic *Methanobrevibacter*,

archaea of the genus *Methanomethylophilus* produce methane from methylated compounds, including methanol[56]. Since methanol is a by-product of methane oxidation by methane monooxygenase, *Candidatus M. alvus* may undermine methanotrophy as an effective means of preventing energy loss to methane emissions from the gut.

Reads which mapped to ciliate reference genomes nearly universally increased in abundance preceding horse death ($\beta = 0.06 \pm 0.02$ SE, $t = 3.144$, $p = 0.002$; Fig. 4B) and showed stronger evidence for multi-year rather than abrupt changes (median $\triangle$AIC = 1.29; Supplementary Fig. 3). Ciliates are associated with methane emissions in ruminants and can facilitate methanogenesis by: (1) cross-feeding methanogens with hydrogen produced by specialized organelles (hydrogenosomes); (2) disrupting non-methane hydrogen sinks by preying on bacteria; or (3) harboring endosymbiotic methanogens[57,58]. Our findings hint that ciliates could similarly promote methanogenesis in the horse hindgut.

A multi-year perspective provides evidence that some mortality-associated changes in the gut microbiome might compound across longer timescales. Features with predicted connections to methanogenesis showed the strongest evidence for multi-year increases in abundance preceding horse death, with some features peaking in abundance years prior to mortality events. But critically, most other features showed stronger evidence for change in abundance only in the year immediately preceding horse death (Supplementary Fig. 3). Mismatches in the timing of features which positively versus negatively predicted horse survival lead us to hypothesize that in some cases, functional changes in the microbiome (e.g. increasing methanogenic energy deficits) might trigger a positive feedback that interferes with a host's ability to regulate their microbiome, thereby causing further deterioration of microbiome function, additional energy loss, and eventually, host death. However, microbiome features strongly associated with horse survival also tended to be more repeatable across years (Fig. 7a), and so could provide a target for selection.

## Discussion

We find evidence that naturally occurring variation in the hindgut microbiome is associated with a host fitness proxy (survival) in a population of feral horses. Our study is observational and so cannot demonstrate causality unequivocally. However, given the reliance of horses on gut microbiota to digest plant material, and because malnutrition is a primary cause of overwinter mortality in this population, we consider causal connections between the gut microbiome and survival to be likely. Analyses of microbe- and microbial gene-specific associations with survival lead us to hypothesize two primary causal mechanisms, pathogenesis and methanogenesis (Supplementary Fig. 4), given that methane emissions can account for a loss of 2–15% of gross energy intake among herbivorous mammals[24–26].

Diet is an environmental factor that likely partially confounded our analyses, since variation in diet characteristics (starch, neutral detergent fiber, protein content, etc.) can affect the gut microbiome and methane emissions[26,28]. However, while diet is frequently emphasized as a primary cause of microbiome variation, studies in lab mice, humans, and wildlife often report surprisingly modest effects of diet[59–61]. In a subset of 85 fecal samples with paired shotgun metagenomic and diet metabarcoding data, we similarly do not find evidence for a relationship between diet composition and gut microbiome variation among Sable Island horses (Supplementary Fig. 5a, b). Furthermore, major diet variation in this population can be largely approximated by including horse location as a covariate within models (Supplementary Fig. 5c, d), due to Sable Island's simple linear geomorphology (Fig. 1). If survival outcomes were caused by diet and correlatively but non-causally linked to methanogenesis, then microbiome features linked to the degradation of plant compounds should have been strongly associated with horse survival. Instead, cellulase gene abundance was not significantly associated with horse survival,

and major plant fiber degrading microbiota in the gut (e.g., Ruminococcaceae and Fibrobacteraceae) had differing relationships with survival. More generally, if diet is an ultimate mechanism that contributes to relationships between the gut microbiome and survival—for example, if malnutrition leaves horses vulnerable to infection or high dietary fiber causes greater methane-related energetic loss—this does not bely microbiome-derived proximate mechanisms as consequential for the host.

Animal methane emissions can be partially affected by diet but are also heritable[62], and microbes linked to methanogenesis (*Methanobrevibacter*, *Christensenella*) have repeatedly been identified as among the most highly heritable gut microbes in human, livestock, and wild animal populations[8,62–64]. Sequential microbiota transplants of *Fibrobacter succinogenes*, *Methanobrevibacter spp.*, *Ruminococcus flavefaciens*, and *Ruminococcus albus* into gnotobiotic lambs experimentally demonstrate that methanogenic outcomes in the gut microbiome are shaped by patterns of colonization and the outcomes of competitive interactions between microbes in the gut[46,47]. Competitive interactions between microbes, which are partly mediated by priority effects and host physiology, are now understood to strongly underlie patterns of microbiome repeatability within individuals through time[65,66]. We observed that the microbiome features strongly associated with survival tended to be more repeatable within individuals across years (Fig. 7a). This result could suggest that, as in other mammals, microbes linked to enteric methanogenesis might have a partially additive genetic basis among Sable Island horses or are influenced by priority effects[67].

Horses produce considerably less methane than expected given their size and dry matter intake, when compared to nearly all other mammalian herbivores[26]. This could indicate that enteric methanogenesis has historically been under strong negative selection in equids. Our findings lead us to suggest that the notable underproduction of methane by equids may be partly attributable to functionally distinct strains of *F. succinogenes* endemic to the horse hindgut[43]. Specifically, we hypothesize two differences that could allow horse-associated *F. succinogenes* strains more effectively mitigate methane emissions than rumen-derived strains—methane monooxygenase and the acetylation of phosphoenolpyruvate carboxykinase[50]. In depth metagenomic, metabolomic, and metaproteomic characterization of in vitro co-cultures of methanogens and horse- versus rumen-specific *F. succinogenes* would be useful in testing these hypotheses[44]. Similarly, direct measures of methane emissions within more tractable wild animal populations, experimental manipulations of enteric methanogenesis using inhibitors or vaccinations[28], and fecal microbiome transplant studies[68] would provide powerful tests for our hypothesis that methane-related energy loss could be a widespread mechanism linking the gut microbiome to fitness variation among wild herbivorous mammals.

Associations between feral horse survival and an expansive shallow shotgun metagenomic dataset of nearly 2400 samples spanning 800 individuals supports the hypothesis that the gut microbiome can contribute to fitness variation within wildlife. These findings from an unmanipulated free-living population are highly complementary to laboratory-based microbiome studies that carefully control for confounds but are far removed from ecologically realistic conditions. Similar studies, conducted across a broader range of environments, hosts, and fitness proxies (e.g., longevity, lifetime reproductive success, offspring recruitment) are critical for broadening our eco-evolutionary understanding of host-associated microbiome variation in nature.

## Methods

### Animal ethics approval

Sample collection and laboratory analyses were performed under Parks Canada Agency Research and Collections Permit SINP-2013-2014, University of Saskatchewan Animal Care Protocol 20090032, and University of Calgary Animal Care Protocol AC18-0078.

## Demographic data collection

The core of the Sable Island Horse long-term individual-based study is comprised of annual population surveys completed during the late summer. All surveys are conducted on-foot by researchers and span a continuous period of 6–7 weeks. In the 2013–2019 timespan represented in the present study, field seasons occurred between July 10th and September 10th. To complete accurate population surveys, Sable Island is subdivided into 7 segments of approximately equal length along its longitudinal axis. These segments correspond approximately to the distance that can be walked by researchers while conducting population surveys within a day. Only a single section is typically surveyed per day, and adjacent sections are not surveyed on consecutive days. Each section is surveyed 6–7 times on average during a single field season.

The location of every horse encountered during population surveys is recorded using handheld GPS devices. Given a history of domestication, and an absence of predators, Sable Island horses are tolerant to the presence of researchers, and most individuals have flight initiation distances of less than 5 meters[69]. Sable Island horse tolerance towards researchers allows us to profile every horse encountered in detail and from multiple angles with Canon Powershot SX530/SX540 cameras. Attention is given to photographing distinguishing physiological features (e.g., dominant coat color, color markings, nasal wrinkles, scarring patterns, chestnut size/shape etc.). Individual horses can be identified across years using these detailed photographs and linked to written observations and location data. Patterns in horse co-occurrence are used to define social band structure, since horses within the same social bands occupy discrete social units on the landscape which are clearly distinguishable during surveys.

Individual-based data allow us to follow every horse in this closed population from birth to death. Using these data, we can estimate horse age to a resolution of one year, since foals and horses ~1 year of age (yearlings) can be easily distinguished from adults. Horses for which a birth year is unknown—i.e., individuals who were older than 1 year of age in the first year of the long-term study (2007)—were assigned an estimated birth date of 2005 (137 of 794 individuals).

## Fecal sample collection

Fecal samples have been collected during targeted fecal sample collection efforts and opportunistically during population surveys, since 2013. Samples are linked to individuals via detailed photographs and data recorded at sampling. All fecal samples are retrieved within 5 min of defecation, and while collecting fecal samples, care is given to avoid portions of feces contaminated with substrate. All samples are collected using clean nitrile gloves, which are turned inside out, sealed, and kept on ice in the field. After a maximum of ~6 h on ice in the field, samples are subset into microcentrifuge tubes (1–2 g of feces per tube), and stored at −20 °C on Sable Island for up to 2 months, before being transferred to long-term storage at −80 °C on the mainland; however, methodological validations in this system show that no difference in microbiome structure can be observed between technical replicates of samples stored at −20 °C versus −80 °C for multiple years[70].

The dataset used in the present study is comprised of 2394 fecal samples collected between 2013 and 2019 from 794 individuals aged 1–14 years. A median of 3 samples were sequenced per individual (min = 1, max = 6), and every individual was represented by a maximum of 1 sample within a given survey year. An additional 73 samples (3% of total sample set) were sequenced but failed to meet minimum sequencing depth thresholds after quality filtering.

## DNA extraction and sequencing

Feces from microcentrifuge tubes (stored at −80 °C) were subsampled into Qiagen 0.7-mm dry garnet bead beating tubes using bleach sterilized metal spatulas (~0.16 g of feces per tube). All samples were kept on ice during and after subsampling to prevent asynchronous thaw prior to DNA extraction. Lysis buffer (680 µl) from the Qiagen QIAamp 96 PowerFecal QIAcube HT kit was added to each tube, and tubes were bead beat in sets of 24 samples for 10 min on a Vortex-Genie 2 using Qiagen's Vortex Adapter (cat. No 13000-V1-24). All subsequent steps conformed to default recommendations from the Qiagen QIAamp 96 PowerFecal QIAcube HT kit single tube start protocol. Final DNA concentrates were eluted in 100 µl of molecular grade water. DNA from 2820 fecal samples across thirty 96-well extraction plates was extracted using this protocol. Each plate comprised of 94 samples, a negative control, and a positive control. Samples were randomized across plates, and randomly distributed within plates. Negative control tubes contained a sliver of clean nitrile glove to mimic possible contamination associated with our feces collection protocol (see above). Positive control tubes contained 75 µl of ZymoBIOMICS Microbial Community Standard II (Log Distribution). Negative and positive controls were processed identically to fecal samples during the DNA extraction protocol. DNA from only a single set of 94 samples was extracted per day, and each extraction set was performed on the same day that samples were weighed to prevent freeze/thaw degradation of samples. DNA elutes were stored at −80 °C prior to quantification, library preparation, and sequencing.

DNA concentration of elutes were quantified using Quant-iT dsDNA Broad Range Assay kits. Samples were concentrated using a centrifugal vacuum concentrator in instances where estimated yields were below 12.5 ng/µl. Notably, all positive and negative controls had values below this detection threshold before and after centrifugal concentration, suggesting nominal kit and glove contamination. DNA extracts were sent to the University of Calgary Center for Health Genomics and Informatics, where shotgun metagenomic libraries were created using a 1:1 mixture of low and high GC primers from the iGenomx Riptide preparation kits (as per manufacturer specifications). Despite falling below input DNA thresholds, positive and negative controls were processed identically to fecal samples. A pilot shotgun metagenomic library of 188 samples (alongside 2 positive controls and 2 negative controls) was sequenced across 2 lanes of an Illumina NovaSeq6000 (300 cycle SP sequencing kit v1.5) to a target depth of 4.1 million 150-bp read pairs per sample. These pilot data were used to validate the use of shallow shotgun metagenomic characterization of the Sable Island horse microbiome[16]. The remaining 2632 samples (alongside 58 positive and 2 negative controls) were sequenced across 4 lanes of an Illumina NovaSeq6000 (300 cycle S4 sequencing kit v1.5) to a target depth 3.8 million 150-bp read pairs per sample (658 samples per lane).

## Bioinformatics

Demultiplexed reads were quality filtered, trimmed, and mapped against the EquCab3 horse reference genome[71] to remove host DNA (<2% in 95% of samples) using the KNEADDATA implementation of BOWTIE2 and TRIMMOMATIC[72–74]. To estimate the taxonomic origins of read pairs, quality filtered reads were translated into all 6 reading frames and mapped to the NCBI BLAST non-redundant protein database using default paired-end implementation of KAIJU at the species/strain level[75]. HUMANN3[73], which uses an implementation of DIAMOND[76], was used to profile the gene family contents of samples against the UniProt50 protein database. In implementing HUMANN3, we bypassed METAPHLAN estimates of taxonomy due to poor representation of Sable Island horse specific strains in available reference databases. As per DIAMOND default settings, we set subject coverage thresholds to 0 to prevent overly conservative filtering of a shallow short read dataset. Summed alignment scores were normalized by gene length and regrouped to MetaCyc reaction gene family groups using HUMANN3[77].

The observation of low DNA extraction yields and sequencing depths from positive and negative controls (when compared to horse fecal samples) suggested negligible kit contamination. Nonetheless, to account for putative contaminant microbiota, we identified and removed taxa which were apparent outliers from abundance-prevalence curves and showed evidence of heterogenous distribution across iGenomx Riptide library preparation plates (Supplementary Fig. 6a-b). Reads which could not be classified to strain or species were binned to the lowest classifiable level, and discarded if they could not be assigned a phylum. Since KAIJU is known to result in a high volume of low abundance false positives[78], taxa not observed at a relative abundance of at least 0.0005 in at least 1 sample were removed (1598 taxa retained). Similarly, gene families not observed at a relative abundance of at least 0.0001 in at least 1 sample were likewise removed (1808 gene families retained). These filtering thresholds also correspond to the abundance at which a shotgun metagenomic dataset of similar depth might be expected to accurately estimate feature abundance[79]. Based on previous comparisons of varied shotgun metagenomic sequencing depths in this system[16], samples which contained fewer than 400,000 read pairs with a taxonomic assignment after filtering were removed prior to analysis. This resulted in the retention of 2394 samples from horses aged 1–14 years.

## Short-chain fatty acid analysis

Forty frozen horse feces stored in 2-ml cryotubes were sent to Microbiome Insights (Richmond, BC, Canada) on dry ice for short-chain fatty acid using gas chromatography-flame ionization detection (GC-FID)[80]. Briefly, 1 gram of feces were resuspended in MilliQ-grade $H_2O$ and homogenized using MP Biomedicals FastPrep lysis system for 1 min at 4.0 m/s. Fecal suspensions were acidified using the addition of 5 M HCl to a pH of 2.0, and after 10 min of incubation at room temperature, centrifuged at 12,298 × g to separate the supernatant. Fecal supernatants were spiked with 2-Ethylbutyric acid for a final concentration of 1 mM. SCFAs (acetic, propionic, butyric, isobutyric, isovaleric, valeric, and hexanoic acids) were detected using gas chromatography (Thermo Trace 1310), coupled to a flame ionization detector (Thermo), and a Thermo TG-WAXMS A GC Column, 30 m, 0.32 mm, 0.25 μm. See Zhao, Nyman, and Jönsson for further details[80]. Succinate was not measured as it is not thought to accumulate in the gut, since it is rapidly converted to propionate[54].

To mitigate the potential confounding effects of host or environmental factors, we constrained our sample selection to samples collected in 2019, from males at least 4 years of age. Samples were selected evenly from horses that lived and those that died during the following winter, and spanned extremes in *Fibrobacterota* relative abundance. Absolute and % SCFA contents, with a focus on acetic, butyric, and propionic acids were modeled as a response to metagenomic feature relative abundance measures (centered and variance standardized), using general liner models. To more directly test our hypothesis that methanogenic energy loss is mediated by microbial competition or facilitation, we also tested for interactive effects between *Methanobrevibacter* (log-transformed relative abundance) and focal metagenomic features in shaping SCFA measures.

A single anomalous sample was excluded from analyses, as it was a statistically significant outlier, and an apparent biological outlier with no detectable butyric acid and 1/5 the acetic acid content of the average sample. Acetic acid was the dominant SCFA (70% ± 4% SD), followed by propionic acid (22% ± 4% SD) and butyric acid (4% ± 1% SD).

## Statistical methods

All analyses were completed in R (v. 4.2.1) primarily using the packages phyloseq[81] and lme4[82]. To estimate the effect of microbiome features on horse over winter survival, we identified environmental and life history variables known to predict Sable Island horse survival, including: sex, age (linear and quadratic terms fitted), sex-specific parental effects (sex:offspring presence in the year of sampling), and mean horse longitude on the island (linear and quadratic terms fitted)[83-85]. The relationships between each composite measure of the microbiome (richness, Shannon diversity, principal components [PC], beta-dispersion) or the abundance of each microbiome feature (centered and variance standardized centered log ratio [CLR] transformed values) on horse overwinter survival were estimated in succession alongside aforementioned covariates within binomial GLMMs (generalized linear mixed models; logit link). Models with principal components included the first 6 PCs simultaneously.

Abundance filtering of taxonomic and gene family estimates was necessary, given the tendency of KAIJU and HUMANN3 to return a high number of low-abundance false positive assignments[73,78]. However, filtering also resulted in severely under-dispersed taxon and gene family richness estimates. Accordingly, taxon and gene richness estimates were obtained from unfiltered and unnormalized data, and GLMMs which modeled horse survival as a response to richness included sequencing depth and log-sequencing depth as covariates to account for unequal sampling effort between samples. Terms for sequencing depth and log-transformed sequencing depth precisely described relationship between sequencing depth and microbiota richness ($R^2 = 0.964$, Supplementary Fig. 7a) and residuals from this sequencing depth-richness relationship are positively correlated with ASV ($r_{pearson} = 0.55$, $t = 5.959$, $p < 0.01$) and genus ($r_{pearson} = 0.49$, $t = 5.142$, $p < 0.01$) richness estimates obtained from paired 16 S rRNA gene amplicon data from biological replicates (Supplementary Fig. 7b).

Year of sample collection was included as a random effect in all models. Inclusion of horse ID to control for repeated measures across years resulted in a singular fit error, since every individual is only represented by a single sample within a given year. Therefore, every individual only has a maximum of one sample collected in the year preceding death. Furthermore, some individuals only have a sample collected in the year immediately preceding their death, while others were not sampled in the year preceding their death. However, analysis of a subset of data which included only a single randomly selected sample per individual yielded the same results (Supplementary Fig. 8). To mitigate sparsity-related inflations of false positive discovery rates in analyses of feature abundance[86], we constrained our analyses to taxon or gene family features which were present in at least 75% of samples (1574 of 1598 taxa, 1416 of 1808 gene families), and samples containing zero reads for a given features were omitted from analyses of that feature. A Benjamini and Hochberg false-discovery rate adjustment was applied to p-values of the same feature type for all models which contained taxon or gene family CLR-transformed read abundances as explanatory variables[20] using the function p.adjust().

To obtain estimates of repeatability for CLR-transformed microbe and gene family abundance estimates, we used restricted maximum likelihood models in ASReml-R[87]. CLR-transformed values were modeled as a response to intrinsic or environmental factors (age as a second order polynomial, longitude as a second order polynomial, collection date, and sex), and microbiome feature repeatability was quantified as the proportion of remaining variance attributable to a random effect of horse ID. In comparing the support for abrupt versus gradual change in the microbiome preceding horse death, we retained only samples collected from individuals who died prior to 2022 (1127 samples, 418 individuals). The CLR-transformed abundances (centered and variance standardized) of each microbiome feature (taxon or gene family) were treated as response variables within GLMMs. We included life history covariates that predicted horse survival (sex, sex:parental status, and linear and quadratic terms for age) and variables predicted to influence microbiome feature abundance (day of year and linear and quadratic terms for longitude of sample collection)[88] as fixed effects within GLMMs. Year and horse ID were included in all models as random effects. For each microbiome feature, we created a pair of models which included the same

covariates, alongside either (a) survival to the following summer as a binary factor (yes or no), or (b) years before death as a continuous numeric variable. To balance sample sizes across years before death estimates, and to prevent disproportionate leverage by samples collected many years preceding horse death, any samples collected more than 5 years preceding death were binned to a ≥ 5 grouping. Both model variants contained the same degrees of freedom, therefore we compared ∆AIC values to infer the relative support for abrupt versus multi-year change in microbiome features preceding horse death. We calculated ∆AIC values as AIC$_{abrupt}$ − AIC$_{multi-year}$, such that negative values indicated greater relative support for abrupt change versus multi-year change, and vice versa. As above, a Benjamini and Hochberg false discovery rate adjustment was applied to the p-values of features with ∆AIC estimates greater than zero. Features identified to show stronger evidence for multi-year change were those which had: (a) a ∆AIC greater than zero, and (b) a statistically significant association with years before death, after Benjamini and Hochberg false discovery rate adjustment.

### Dietary metabarcoding analysis

To characterize variation in Sable Island horse diets, we obtained dietary metabarcoding data from a subset of 85 fecal samples. All samples were collected from female horses in 2014 and were paired with shotgun metagenomic sequencing data derived from the same fecal DNA extracts. Briefly, we PCR amplified the trnL-UAA P6 loop chloroplast gene region using g region 5′-GGGCAATCCTGAGCCAA-3′ and h region 5′-CCATTGAGTCTCTGCACCTATC-3′ primers[89]. Both primers consisted of a pool of 7 salt-free primers mixed in equal amounts containing between one and seven random bases (N) at the 5′ end to increase diversity of amplicons and an adapter sequence to allow incorporation of barcodes in a second PCR.

Prior to conducting PCR amplification on the entire set of samples, we first identified suitable PCR conditions by comparing amplification using the conditions detailed below but using either 40 ng or 80 ng of template, 25 or 30 amplification cycles, and an annealing temperature of 60 °C or 62 °C. Based on visualization on 2% agarose gels, we determined that using 40 ng of template, 30 amplification cycles, and an annealing temperature of 62 °C performed best among those conditions. Co-loading of a 25-766 bp ladder (Quick-Load Low Molecular Weight DNA Ladder, New England Biolabs) was also used to determine that most amplicons (including primer sequences) were approximately 200 bp long. A set of 95 samples (which included 85 samples with paired metagenomic data) and 1 negative control (molecular grade water) were then amplified using PCR with the following reagents: 5 µL 5X KAPA HiFi HotStart Fidelity Buffer (KAPA Biosystems, USA), 0.75 µL g primer (10 µM), 0.75 µL h primer (10 µM), 0.75 µL dNTPs (10 mM), 0.5 µL KAPA HiFi HotStart Polymerase (0.5 U), 9.25 µL ddH$_2$O, and 8 µL template DNA (standardized to 10 ng/ul). Thermocycling conditions were 95 °C for 2 min, followed by 30 cycles of 98 °C for 20 s, 62 °C for 15 s, 72 °C for 15 s, followed by a 2-min final extension at 72 °C.

After confirming successful PCR amplification by visualizing 5 µl of product on a 2% agarose gel, the remainder was purified using AMPure XP Magnetic Beads following manufacturer protocols with a 1.8X bead to template ratio (Beckman Coulter, Inc.). Ninety-six pairs of Unique Dual Indices (UDI, IDT for Illumina Set A) and P5/P7 sequencing tags were then added to the amplicons in a second PCR amplification. Reagents for this second PCR consisted of: 5 µL 5X KAPA HiFi HotStart Fidelity Buffer, 1.25 µL Forward Primer (10 µM), 1.25 µL Reverse Primer (10 µM), 0.75 µL dNTPs (10 mM), 0.5 µL KAPA HiFi Polymerase (0.5 U), 10.25 µL H$_2$O, and 6 µL of purified first round PCR product. Thermocycling conditions were 98 °C for 45 s, followed by seven cycles of 98 °C for 20 s, 63 °C for 20 s, and 72 °C for two minutes. Products were again purified using AMPure XP magnetic beads (1.8X), visualized on a 2% agarose gel, and quantified using the Quant-it BR kit (2 µl of template).

Samples were then pooled in equal amounts (100 ng/sample) into a single library, except for the negative control for which 5.2 µl was added (which corresponded to the average volume added per sample). This library was quantified using absorbance with a Take3 Trio microplate adapter (averaged from 8 2 µl replicates), diluted to ~5 ng/µl using molecular grade water, and re-quantified using a Quant-it HS kit (10 µl of template, 8 replicates). The library concentration in nM was then estimated assuming an average library size of 242 bp (inferred from the dominant peak on an Agilent TapeStation D1000 ScreenTape) and diluted to 1 nM using 10 mM Tris-HCl (pH 8.5). This 1 nM library was further diluted to 50 pM using Tris-HCl and combined 20:1 with 50 pM PhiX Control v3 (Illumina, San Diego, USA). The library (20 µl input) was sequenced on a local Illumina iSeq 100 with a v1 300-cycle Reagent Kit following the iSeq 100 Sequencing Guide, and demultiplexed FASTQ files created using the Local Run Manager Generate FASTQ Analysis Module V2.0.1.

Raw reads were trimmed, quality controlled and merged using DADA2 in R[90]. We used a workflow developed for amplicon sequencing of the ITS2 gene region modified as per Poissant et al.[91]. A custom trnL reference database was constructed using sequences obtained from the NCBI SRA. This database contained reference sequences from 153 plant species (or close relatives) identified during detailed plant surveys conducted in 2019 and 2021. Reference sequences obtained from closely related plant species were used in instances where an exact species match was not available. Amplicon sequence variants (ASVs) were assigned to the finest taxonomic scale possible using the assignTaxonomy() function and an 80% confidence threshold cutoff[92].

We analyzed dietary metabarcoding profiles at the levels of genus and order, to allow for the possibility that plant nutritional profiles could be either deeply or shallowly phylogenetically conserved. ASVs which could not be assigned to a genus or order were instead binned to their finest classifiable grouping.

No evidence for a relationship between gut microbiome and diet dissimilarity (Aitchison distance) was observed among mantel tests, regardless of whether we grouped diet derived ASVs to the level of genus ($p = 0.55$; Supplementary Fig. 5a) or order ($p = 0.28$; Supplementary Fig. 5b). The absence of a strong relationship between gut microbiome variation and the Sable Island horse diet is consistent with previous phylogenetic null modeling results from this system, which more strongly indicate microbial dispersal limitation (rather than heterogenous selection due to diet) as a key determinant of microbiome variation in this population[88].

At both the level of genus and order, the first principal component of Aitchison distances in diet dissimilarity was strongly quadratically associated with longitude, whereby by fecal samples collected from the eastern and western extremes of Sable Island tended to differ from samples collected towards the island midpoint (genus: $\beta_{linear} = 4.72$, $t_{linear} = 7.553$, $p_{linear} < 0.01$; $\beta_{quadratic} = -5.445$, $t_{quadratic} = -8.709$, $p_{quadratic} < 0.01$; order: $\beta_{linear} = 5.30$, $t_{linear} = 8.856$, $p_{linear} < 0.01$; $\beta_{quadratic} = -5.16$, $t_{quadratic} = -8.618$, $p_{quadratic} < 0.01$; Supplementary Fig. 5c, d). Fitting longitude as a second-order polynomial within survival models is therefore expected to control for the effects of major dietary variation on horse survival.

Longitudinal differences in horse diet are likely strongly attributable to sea sandwort (*Honckenya peploides*) at the eastern and western tips of Sable Island (Supplementary Note 2)[16,88]. Sea sandwort is lower in fiber (49% neutral detergent fiber) than marram grass (*Calamagrostis breviligulata*), which comprises a large part of the Sable Island horse diet (63% neutral detergent fiber). However, the exclusion of samples collected from horses with access to sea sandwort from survival analyses (exclusion of 325 samples with a longitude less than −60.08°; retention of 2069 samples from 747 individuals) had no effect on the estimated relationships between microbiota and horse survival (Supplementary Fig. 9). The absence of strong relationships between diet and microbiome dissimilarity, the robustness of our

findings to the exclusion of fecal samples likely to represent the most divergent diets, and evidence for inter-annual repeatability of the gut microbiome within individuals (Fig. 7a) do not support the interpretation that the associations we observed between the gut microbiome and horse survival are caused by diet.

## Reporting summary

Further information on research design is available in the Nature Portfolio Reporting Summary linked to this article.

## Data availability

The shotgun metagenomic sequence and dietary metabarcoding sequence data that support the findings of this study have been deposited in the NCBI SRA under the BioProject accession codes PRJNA1102860, PRJNA880353, and PRJNA1104620. GC-FID SCFA data and sample metadata has been deposited on FigShare (https://doi.org/10.6084/m9.figshare.25676670). Source data is provided as a source data file.

## Code availability

All code has been deposited on FigShare (https://doi.org/10.6084/m9.figshare.25676670).

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

## Acknowledgements

We thank the numerous students, research assistants and volunteers who have contributed to data and sample collection and processing. We also thank the University of Calgary Center for Health Genomics and Informatics for assistance with library preparation and sequencing. In-kind and logistical support was provided by the Department of Fisheries and Oceans Canada (DFO), Canada Coast Guard, the Bedford Institute of Oceanography (DFO Science), Environment Canada, Parks Canada Agency, Maritime Air Charters Limited (Sable Aviation), and Sable Island Station (Meteorological Service of Canada). This work was funded by a Margaret Gunn Endowment for Animal Research Grant (JP), University of Calgary Research Grants Committee Seed Grant (JP), Heather Ryan and L. David Dubé Veterinary Health and Research Fund Grant (PDM, JP), Morris Animal Foundation D20EQ-05 (JP, AJW, PDM), Natural Sciences and Engineering Research Council of Canada Discovery Grant 2019-04388 (JP), Natural Sciences and Engineering Research Council of Canada Discovery Grant 2016-06459 (PDM), Canada Foundation for Innovation Leaders Opportunity Grant 25046 (PDM), Natural Sciences and Engineering Research Council of Canada Vanier Scholarship (MRS), Alberta Innovates Graduate Student Scholarship (MRS), and a Izaak Walton Killam Pre-Doctoral Scholarship (MRS).

## Author contributions

Conceptualization: M.R.S, P.D.M., A.J.W., J.P., Data curation: M.R.S., P.D.M., S.M., R.J.G., J.P., Formal Analysis: M.R.S., A.J.W., R.J.G., J.P., Funding acquisition: M.R.S., P.D.M., A.J.W., J.P., Investigation: M.R.S., P.D.M., S.M., A.W.P., J.P., Methodology: M.R.S., P.D.M., S.M., R.J.G., A.J.W., J.P., Project administration: P.D.M., J.P., Supervision: P.D.M., J.P., Visualization: M.R.S., Writing—original draft: M.R.S., Writing—review & editing: M.R.S., P.D.M., S.M., R.J.G., A.J.W., J.P.

## Competing interests

The authors declare no competing interests.
