## [Peer Review File · Nature Communications]

Methanogenic patterns in the gut microbiome are associated with survival in a population of feral horsesREVIEWER COMMENTS

Reviewer #1 (Remarks to the Author):

Dear editors and authors,

Thank you for the opportunity to review this interesting manuscript. This is undoubtedly the largest study in this field using almost 2400 fecal samples from wild horses analyzed by shotgun sequencing. Samples were collected over a 6 years period constituting an unique model of an environment with high selection pressure.

The manuscript is very well written, the analyzes are sound and conclusions supported by their results. I have only minor suggestions for the authors consideration.

Title: why to use "a wild herbivore" if you are evaluating horses? I would specify that in the title.

It would be helpful maybe having a graphical summary of the rationale on how methane consumption by the microbiota can predict survival. You could include other main results as the increase/decrease of certain microorganisms predicting mortality.

The discussion about causality is mentioned in the conclusion session, but I believe this needs to be further expanded. Although authors used their words carefully, the manuscript gives the impression that increased methane production is responsible for death. Other unaccounted variable should be mentioned. It is possible that the changes in the microbiome of non-survivors were actually caused by undelaying disease, inflammation or other factors. For instance, dental problems could difficult mastication that could change digestibility and therefore microbiome composition, leading to increased features linked to methanogenesis.

Other limitations to be discussed: lack of metabolites and gene expression analysis.

Line 36: can 6 years be considered "long term" for a selection study?

Line 122-123: I found this statement not very appropriate. The common hypothesis is not that high diversity is associated with survival or longevity, so your results are not contradictory to the current hypothesis. You should consider to discuss that low diversity is a common feature of food animals treated with low doses antibiotics, which is also associated with weight gain through microbiome selection: Cho et al., Nature. 2012;488(7413):621-6. doi: 10.1038/nature11400.

Line 167: I don't agree that Clostridia sp. are listed as opportunistic pathogens. Only a minority of the hundreds of species in that genus are actually pathogens, but the vast majority are commensal. In fact, those commensals are present in high abundances in healthy horses.

Line 262: it is not clear why you started the sentence with "Rather than cellulase or urease,".

Line 280: why did you keep the strongest association to the end? It would make sense to start with your major findings, unless they are less important in your data interpretation?

Line 354-358: this is a strong statement. Could you elaborate a little more and maybe use data from your results to better support this hypothesis?

Line 366-367: how do you intend to do that? Can you give some examples? I would say that the next step would be to test your hypothesis in an experimental model.

It is very difficult to read the legends in pretty much all the figures. For figure 2, you should write *Fibrobacter succinogenes* the first time it appears in the text.

Reviewer #2 (Remarks to the Author):

Overall, the manuscript is well written and very clearly presented. It was easy to follow the different storylines, which can be tricky to accomplish when presenting so much data. The figures made the results easy to understand, and I appreciated the historical information provided in the supplemental material to understand the life history of these horses and how their microbiomes were affected by their ecosystem.

I couldn't find anything to suggest adding or fixing - well done!

Reviewer #3 (Remarks to the Author):

The current paper represents a longitudinal study of the gut microbiota in horses aiming to link changes in the gut microbiota with the overwinter survival in wild horses. This experiment implies a substantial amount of field work to collect the biological samples and environmental information, as well as for the sample analysis and interpretation.

However, the experimental design may be insufficient to test the hypothesis as the gut samples were collected nearly a year before the potential death of the horses, therefore the data interpretation may be biased as there are a large number of environmental, health or behavioural factors across horses that can bias the results. Moreover, correlation does not imply causality.

Title should say "horse" instead of "herbivores" as these findings may not apply to other animal species. Moreover, "predict" should be turned down to "may be linked".

L510 Explain why Principal Component (PC) Analysis was performed using the first 6 PC instead of using other plotting analysis which includes the entire variability (CCA, PCoA, etc.)

L514 Differences in sequencing depth lead to large bias in diversity index. Thus, it is necessary to explain why diversity was calculated before normalization. Using sequencing depth as covariant may not be sufficient to address this issue and needs further justification.

-Gut microbiota can change substantially from one year to another mostly depending on the diet. Therefore, analysis of the gut microbiota one year before death (or even more) cannot be considered as representative description of the gut microbial ecosystem to establish correlation or causality of the horse fitness.

L530 "In relation to potential abrupt vs gradual changes in the microbiome preceding horse death" It seems unlikely to make associations between changes in the gut microbiota and horse death considering that many samples were collected months or even years before the horse death. Similarly, some horses could have been sampled when the horse was suffering a disease process but which did not end up in the horse death. All these situations indicate the presence of many experimental bias which can limit the possibility of drawing robust conclusion.

It should be identified the reason of each horse deaths in order to avoid potential bias. For example, a lack of gut microbial disbiosis would be expected if a horse die due to an accident.

L124 All herbivores prefer eating highly digestible forage instead of highly fibrous forage if they have the chance, as this imply a lower digestive effort. In ruminants a decrease in gut diversity is observed when animals consume highly digestible diets. Therefore, this may indicate that horses with lower diversity may eat the best grass (more digestible) and ultimately have a higher overwinter survival. On the contrary, the weak horses may have problems to reach the best grass (due to walking or group hierarchy limitations). This observation would explain why horses with a higher survival rate have a lower diversity and abundance of methanogens and ciliates (which are associated to de digestion of fibrous feeds), whereas horses with a lower survival rate seems to be more adapted to eat fibrous diet (possibly because they are unable to reach the best grass). If this is true, the differences in the microbiota must be considered as a consequence of the diet but not the reason to explain the horse survival.

L132 A deviation of an individual's microbiome composition from the population's average does not necessarily imply a dysbiosis. It may indicate an adaptation to a specific environment or diet.

L137-140 It is well known that the diet is the main driver that determines the gut microbiota being much more relevant than other minor factors such as the geographical location, host genetics or even the animal species (a horse and a zebra eating the same diet tend to have a similar microbiota). Thus, these associations do not imply causality as they may be mediated by the type of diet consumed by the horse.

L148 Can the AIC model be overfitted? It is well known that prediction models with a reasonable accuracy can be developed when using an internal validation process which implies the use of the same dataset to develop the model and to test its performance. On the contrary, an external validation, which implies the use of a large proportion of samples for model development (e.g. 90%) and the remaining samples for model validation, would imply a more robust determination of the model performance and it would prevent model overfitting (Fig S1). The inclusion of indicators of the model performance such as the determination coefficients of calibration R^2_C , cross validation R^2_{CV} , prediction R^2_P and their respective root mean square errors (RMSEC, RMSECV, RMSEP), the performance to prediction ratio (RPD) or the range in reference ratio (RER) would be appreciated.

L171-179 It is mentioned that food limitation is the most common cause of mortality in Sable Island horses. Thus, changes in the gut microbiota are most likely associated to insufficient feed intake or to the intake of low quality diet (fibrous diet) which is often associated to high abundance of methanogens and methanogenesis. Thus, changes in the gut microbiota cannot be considered as the cause of higher or lower survival as they are likely an indirect effect derived from other aspects such as physical, health or social horse limitations which can limit food intake or lead to changes in the diet (e.g. more fibrous diet). A chemical analysis of the horse faeces (e.g. fiber and protein content) and/or in vitro quantification of the CH_4 production after incubation of horse faeces with the same feed substrate across animals could help to prove this hypothesis. Alternatively, a quantification of the fermentation products (e.g. volatile fatty acids) from the horse faeces could provide useful information to link gut microbiota and function.

L198-208 When analysing metagenomics data, it is very common to find associations between gut microbes and genes as huge number of combinations are tested. However, those associations do not imply causality. This is particularly critical when assuming microbial effects on gut fermentation or CH_4 production without direct measurements of those parameters. Moreover, most of the concepts described here (H_2 sinks, sulphate or nitrate reduction) are derived from ruminant studies which have a different digestive physiology as they are foregut fermenters while horses are hindgut fermenters.

L212-226. Again, CH_4 , propionate, lactate and fermentation products were not measured in this study. As a result, this paragraph is highly speculative.

L228 *Fibrobacter succinogenes* produces acetate and succinate as main fermentation products. As acetate production releases the highest amount of H_2 which is transformed into CH_4 by methanogens, it may imply an overall increase in CH_4 production. As indicated before, it has been observed that *F. succinogenes* is one of the most fibrolytic microbes in the gut and its abundance is linked to the fibre intake. This again, highlights that the diet composition may be the key driver in the overwinter horse survival, while the effect in the gut microbiota may be indirect.

L274-277 This sentence is highly speculative as fermentation products were not measured in this study.

Tables S4 to S7 are not provided.

L342 The increase of protozoa during the year before death may indicate food scarcity which forced that horse to eat a high-fibrous diet as protozoa are fibrolytic microbes. This highlights that the type of diet consumed by the horse is linked to the survival rate, possibly as less fitted horses are pushed into the worst pastures with highly fibrous grasses.

L361 It should be defined what do you mean by "natural occurring variations in the hindgut microbiota". For example, a lame or a nearly-blind or shy horse can have a different rumen gut microbiota in comparison with the population average, but this is driven by the diet as they are less fit and likely pushed to areas with the low-quality grass.

L365 "causal dependence of survival on methanogenic pattern" can be a speculation as the CH₄ production was not measured and there are potential bias data interpretation.

L374-376 Most of the CH₄ production from livestock is coming from ruminants, therefore the horse, as a hindgut fermenter, cannot be used as an accurate model for ruminants.

RESPONSE TO REVIEWER COMMENTS

Reviewer #1 (Remarks to the Author):

Dear editors and authors,

Thank you for the opportunity to review this interesting manuscript. This is undoubtedly the largest study in this field using almost 2400 fecal samples from wild horses analyzed by shotgun sequencing. Samples were collected over a 6 years period constituting an unique model of an environment with high selection pressure.

The manuscript is very well written, the analyzes are sound and conclusions supported by their results. I have only minor suggestions for the authors consideration.

>>We thank the reviewer for their positive feedback, and their suggestions which have helped to improve our manuscript.

- **Title: why to use “a wild herbivore” if you are evaluating horses? I would specify that in the title.**

>>As the reviewer suggests, we now specify ‘feral population of horses’ in the title. We initially used the term ‘wild herbivore’, as we wanted to keep the title as broad as possible to ensure it finds its target audience (wildlife ecology/evolution researchers), who might otherwise overlook a paper on horses, if they think it pertains to domestic populations. We feel the specification of ‘feral’ will help to accomplish this objective.

- **It would be helpful maybe having a graphical summary of the rationale on how methane consumption by the microbiota can predict survival. You could include other main results as the increase/decrease of certain microorganisms predicting mortality.**

>>As suggested by the reviewer, we now include a graphical summary in the supplemental materials which outlines our hypotheses of how 19 different survival-associated metagenomic features discussed in the main text are connected to methanogenesis. (Supplementary Figure 4).

- **The discussion about causality is mentioned in the conclusion session, but I believe this needs to be further expanded. Although authors used their words carefully, the manuscript gives the impression that increased methane production is responsible for death. Other unaccounted variable should be mentioned. It is possible that the changes in the microbiome of non-survivors were actually caused by undelaying disease, inflammation or other factors. For instance, dental problems could difficult mastication that**

could change digestibility and therefore microbiome composition, leading to increased features linked to methanogenesis.

>>As suggested by the reviewer, we have expanded upon our discussion of causality in the patterns we observed (L516, L522, L562). The reviewer's point about inflammation or dental problems causing microbiome disruption is a good one, however, we would argue that if inflammation or other physiological problems indeed contribute to horse death, this would not mean that microbiome disruptions aren't a proximate cause of mortality, as now acknowledged on L536. We now also supplement our previous analyses with dietary metabarcoding data which do not support the existence of a strong relationship between gut microbiome and diet variation (L526), and report that survival-associated microbiome features are repeatable within individuals across years (L529)

- **Other limitations to be discussed: lack of metabolites and gene expression analysis.**

>>As suggested by the reviewer, we generated a complementary short-chain fatty acid dataset from a subset of samples, which we use in our revised manuscript to test the predictions derived from our metagenomics dataset. Analyses of these data support our predictions (L279, L381, L457, L466), and so we thank the reviewer for their recommendation which we feel has improved the quality of our manuscript.

We agree with the reviewer that gene expression data would also be useful for testing these predictions. However, we were unable to also generate a transcriptomic dataset, given finite time and financial resources, as well as the likely unsuitability of archived fecal samples for quality RNA recovery. We agree that this would be an interesting next step and test of our predictions, and future samples will be collected with these analyses in mind. We now highlight on lines L377 & L560 that more comprehensive 'omics data would be useful for testing the hypotheses we outline.

- **Line 36: can 6 years be considered “long term” for a selection study?**

>> Our fecal sample collections spanned 7 years of collection, which is relatively long-term compared to most other studies of wildlife microbiomes. However, here our use of 'long-term' was in more direct reference to the broader Sable Island Horse data, which began collections in 2007. Data collected since 2007 have been critical to our analyses (for example, providing us measurements of horse age and an understanding of the ecology of this population, which has helped us identify potential confounds). We have amended wording to make this clearer (L97).

- **Line 122-123: I found this statement not very appropriate. The common hypothesis is not that high diversity is associated with survival or longevity, so your results are not contradictory to the current hypothesis. You should consider to discuss that low diversity is a common feature of food animals treated with low doses antibiotics, which is also associated**

with weight gain through microbiome selection: Cho et al., Nature. 2012;488(7413):621-6. doi: 10.1038/nature11400.

>> We wholeheartedly agree with the reviewer that high diversity is not always associated with survival, longevity, or other host performance metrics. Unfortunately—especially within the wildlife microbiome literature—high diversity is commonly presumed to be beneficial, and so we felt it important to flag this common (if misguided) hypothesis for our target audience (which may include wildlife researchers new to the world of microbiome research).

In response to the reviewer's concern, we have amended our wording to make it clear we do not ourselves hypothesize that microbiome richness is beneficial to the host (L137). We now also cite two perspectives/review articles on this topic, which we feel provide a good overview of how nuance is required when interpreting patterns of alpha diversity. The reviewer suggested citation is indeed exemplary of the fact that alpha diversity is not always best, but since we are at our citation limit, we feel the two reviews now cited provide a more fulsome overview of this topic.

Line 167: I don't agree that Clostridia sp. are listed as opportunistic pathogens. Only a minority of the hundreds of species in that genus are actually pathogens, but the vast majority are commensal. In fact, those commensals are present in high abundances in healthy horses.

>>We thank the reviewer for this comment. We meant '*Clostridium*' rather than 'Clostridia'; this was a typo which we have now corrected. We further recognize that not all *Clostridium* are necessarily pathogens or might not be pathogenic in all contexts. However, *Clostridium* are frequently associated with disease states in horses, and so we feel it important to highlight this genus. In interpreting our results, we have tried to strike a balance between acknowledging the potential for pathogenesis, versus making broad strokes claims about pathogenicity of various microbes, which is often highly context dependent.

- **Line 262: it is not clear why you started the sentence with “Rather than cellulase or urease,”.**

>>As suggested by the reviewer, we have removed this phrasing which was meant as a bridge to the end of the preceding paragraph. We have reworded for clarity (L347).

- **Line 280: why did you keep the strongest association to the end? It would make sense to start with your major findings, unless they are less important in your data interpretation?**

>>The reviewer's suggestion is a valid alternative narrative framing. Ultimately, this result was placed at the end of this section, so that we could unpack results in a more organized manner, without overwhelming the reader with too much information from the beginning. For example, placing the result pertaining to methane monooxygenase at the start of the results section would require simultaneously presenting gene family and

taxonomic results, discussing strain-specific effects, while also explaining the biological significance of methanogenesis without providing the reader the context necessary to understand the broader picture. We hope that in its current iteration, we are better able to describe how our results fit together to provide the basis for a key methane-related hypothesis.

- **Line 354-358: this is a strong statement. Could you elaborate a little more and maybe use data from your results to better support this hypothesis?**

>>L359: As suggested by the reviewer, we have clarified this statement, specifying that most features did not show strong evidence of multi-year change (including those positively associated with survival, as well as some features negatively associated with survival; L504). We have now also expanded this section to include analyses of microbiome repeatability which indicate that features strongly connected to horse survival also tended to be more repeatable within individuals across years (L429).

- **Line 366-367: how do you intend to do that? Can you give some examples? I would say that the next step would be to test your hypothesis in an experimental model.**

>>As suggested by the reviewer, we now include examples of next steps which could be used to test the hypotheses we propose (L562).

- **It is very difficult to read the legends in pretty much all the figures. For figure 2, you should write *Fibrobacter succinogenes* the first time it appears in the text.**

>>As suggested by the reviewer, we have provided a new set of figures with larger and clearer text.

Reviewer #2 (Remarks to the Author):

Overall, the manuscript is well written and very clearly presented. It was easy to follow the different storylines, which can be tricky to accomplish when presenting so much data. The figures made the results easy to understand, and I appreciated the historical information provided in the supplemental material to understand the life history of these horses and how their microbiomes were affected by their ecosystem.

I couldn't find anything to suggest adding or fixing - well done!

>>We thank the reviewer for their time and positive feedback. We also appreciate the confirmation about the clarity of our interpretations.

Reviewer #3 (Remarks to the Author):

The current paper represents a longitudinal study of the gut microbiota in horses aiming to link changes in the gut microbiota with the overwinter survival in wild horses. This experiment implies a substantial amount of field work to collect the biological samples and environmental information, as well as for the sample analysis and interpretation.

>>We thank the reviewer for their time and thoughtful feedback which we feel have helped to strengthen our manuscript. We also thank the reviewer for their acknowledgement of the substantive time investment that contributed to the construction of the expansive dataset used in this research.

- **However, the experimental design may be insufficient to test the hypothesis as the gut samples were collected nearly a year before the potential death of the horses, therefore the data interpretation may be biased as there are a large number of environmental, health or behavioural factors across horses that can bias the results. Moreover, correlation does not imply causality.**

>>We agree with the reviewer that correlation does not imply causation but argue that correlational observations can nonetheless be used as the basis to form testable hypotheses. Observation is a critical first step in the scientific method.

We also note that all long-term individual-based ecological study are complicated by environmental, health, and behavioural factors. This is in fact integral to the entire premise of long-term ecological research, which seek to parse biological patterns in complex real-world ecological systems, and in doing so, garner a greater fundamental understanding of nature. Despite their inherent complexity, long-term individual-based study systems have disproportionately contributed to our understanding of ecology and evolutionary biology (Clutton-Brock and Sheldon 2010, *Science*; Clutton-Brock and Sheldon 2010, *Trends in Ecology and Evolution*). While valuable, designing an artificial system which controls for these factors would defeat the purpose of our research, which was to determine whether we could detect associations between the microbiome and a host fitness proxy in the wild.

There is in fact an abundance of laboratory experiments which demonstrate the importance of microbiome variation under highly controlled conditions—whether these signals are detectable in under real-world conditions is less clear and this is precisely the knowledge gap we sought to fill. Nonetheless, we note that otherwise confounding environmental gradients on Sable Island can be effectively controlled for by measurements of longitude, given Sable Island's simple linear geomorphology (1.5 km x 45 km; Figure 1). This is a major advantage to our study system, compared to nearly all other wild animal study systems.

Although samples were collected months rather than a year prior to horse death, as now clarified on L108, we hypothesize that it is gut microbiome variation during the summer that is critical for overwinter horse survival, since Sable Island horses are dependent upon energy reserves amassed during the summer to live through the winter (when there is little quality plant biomass available). It is therefore digestive efficiency during these narrow windows of quality forage availability that are critical for horses. This hypothesis appears to be supported by our results.

With respect to the reviewer's concerns that microbiomes sampled months prior to horse death are irrelevant to survival, recent research suggests that, despite the ecological complexities of studying microbiome variation in wild animal populations, gut microbiome variation is highly repeatable within individuals across years of sampling (Björk et al. 2022, Nature Ecology & Evolution). To address this point, we now provide estimates of repeatability across the 7 years of sampling (L429). As in other recent studies, we report that microbiome features are indeed repeatable across years, but also that features most strongly connected to horse survival are those which are most repeatable.

- **Title should say “horse” instead of “herbivores” as these findings may not apply to other animal species. Moreover, “predict” should be turned down to “may be linked”.**

>> As suggested by the reviewer, we have changed our title to “associated with”. We had not intended to imply causation with the word ‘predict’ but understand the reviewers concerns that this might be misinterpreted by readers. Likewise, we have changed “herbivores” to “feral horse population”. We initially sought to keep our title broad, to ensure this manuscript finds its target audience (wildlife ecology and evolutionary biology researchers) and to not mislead researchers who study domestic horses. We feel the specification of ‘feral’ will help to accomplish these objectives.

- **L510 Explain why Principal Component (PC) Analysis was performed using the first 6 PC instead of using other plotting analysis which includes the entire variability (CCA, PCoA, etc.)**

>> We broadly sought to test whether microbiome data was informative of horse survival, above and beyond environmental variables or host-level factors. A PERMANOVA analysis, which uses beta diversity matrices as a response variable, does not accomplish this objective. In parameterizing generalized linear mixed effects models, it was necessary to decompose multi-dimensional microbe community and gene family composition data into univariate measures (PCs) that could be fitted as fixed effects within our model competition framework. Of course, the Principal Component Analysis we performed partitions complex community data across $n-1$ dimensions (where n = the # of taxa / gene families). It was therefore also necessary to use only a subset of PCs, rather than attempting to fit thousands of PCs within a single model.

The plotting analyses the reviewer mentioned can be a useful visualization tool, but we argue it is not overly informative nor appropriate for our research question, in which horse survival (rather than microbiome composition) is the focal response variable. We do indeed find among PERMANOVA analyses that microbiome community/gene family composition significantly differs between horses that survived versus died (even after controlling for environmental and life history factors). However, this does not provide information about what dimensions of the microbiome differ between groups, nor does it tell us whether a null 'non-microbiome' model would be more parsimonious. This is why we made the decision to adopt an AIC model competition framework using a subset of the top PCs.

- **L514 Differences in sequencing depth lead to large bias in diversity index. Thus, it is necessary to explain why diversity was calculated before normalization. Using sequencing depth as covariant may not be sufficient to address this issue and needs further justification.**

>> As suggested by the reviewer, we now detail on L792 and Supplementary Figure 7 that 'depth' & 'log(depth)' fit as covariates very precisely describes the relationship between sequencing depth and richness (Supplementary Figure 7; $R^2 = 0.964$). In our survival analyses, we are thus testing whether microbiomes being more or less diverse than expected given their sequencing depth is predictive of horse survival. We further demonstrate these estimates of richness are positively associated with richness estimated from more conventional rarefied 16S rRNA data, and thus provide biologically relevant estimates of alpha diversity. As explained in the Methods section (L789) we opted for this alternative method of richness estimation because conventional rarefaction methods resulted in estimates that were severely under-dispersed, which is likely a result of the economical shallow shotgun metagenomic sequencing method used in our study.

- **Gut microbiota can change substantially from one year to another mostly depending on the diet. Therefore, analysis of the gut microbiota one year before death (or even more) cannot be considered as representative description of the gut microbial ecosystem to establish correlation or causality of the horse fitness.**

>> To address the reviewer's valid concern about microbiome repeatability, we now include analyses demonstrating that microbiome features are repeatable within individuals across the 7 years of sampling, and that features which are the most repeatable are those most strongly associated with horse survival (L429). These new results do not support the interpretation that the survival-associations we observed are transient and primarily attributable to diet.

The microbiome repeatability we report are aligned with the latest research, which demonstrates that, while microbiomes can be dynamic, they are also idiosyncratic and highly repeatable within individuals across years (Björk et al. 2022, Nature Ecology

& Evolution; Valles-Colomer et al. 2023, Nature; Zhou et al. 2024, Cell Host & Microbe). Similarly, research in human, livestock, and wildlife populations also find evidence for heritability (Goodrich et al., 2016, Science; Grieneisen et al., 2021, Science; Ryu & Davenport, 2022, Annual Review of Animal Biosciences; Wallace et al., 2019, Science Advances). One of the largest microbiome studies to date (>8000 human subjects and 100s of covariates) finds that diet accounts for ~1% of total variation in the microbiome, which is ~<1/20th the effect size of cohabitation and additive genetic effects estimated within the same study (Gacesa et al. 2022, Nature).

Nonetheless, as the reviewer notes, we agree that diet can contribute to variation in the microbiome. To more directly investigate the concerns raised by the reviewer, we thought it would be informative to test for connections between the microbiome and horse diet. To that end, thanks to the reviewer's feedback, we now provide dietary metabarcoding data. Analysis of these data find no correlations between microbiome and diet dissimilarity among Sable Island horses (L526; Supplementary Materials; Supplementary Figure 5a,b). Furthermore, we find that dietary variation can be largely accounted for by fitting longitude as a covariate within survival models (Supplementary Figure 5c,d). These results are consistent with our previous work, wherein using phylogenetic null models we find that dietary factors do not appear to be the main contributor to between-individual microbiome differences (Stothart et al. 2021; Molecular Ecology). To further test the robustness of our findings, we also performed an additional set of selection analyses, after excluding horses from the western tip of Sable Island which have the most strongly differentiated diet profiles (attributable primarily to access to a low-fibre forage, sea sandwort; Supplementary Materials). Exclusion of these putative dietary outliers had negligible effects on the estimated associations between microbiome features and horse survival (Supplementary Figure 9). We further now clarify in the main text:

1. If the associations we observed between the microbiome and horse survival were proximately attributable to diet, then we would have expected microbiota and microbial gene contents related to the degradation of plant fibre to be strongly related to survival. Instead, cellulase gene abundance was not significantly associated with survival and major fibrolytic microbiota (Ruminococcaceae and Fibrobacteraceae) showed divergent associations with horse survival. Similarly, many asaccharolytic protein-degrading microbiota exhibited some of the strongest negative associations with horse survival. (L533).
2. More so than the plant species composition of horse diets (as measured with metabarcoding data), forage fibre content is strongly expected to increase across the growing season. However, horses which survived versus died were randomly sampled between July to September. If dietary fibre was the causal mechanism underlying survival, and the microbiome was only correlatively but non-causally linked to survival, we would expect seasonal decreases in forage quality to have obscured our ability to estimate selection acting on the microbiome (rather than cause spurious associations as the reviewer suggests).

3. Even if diet variation partly underlies variation in methane emissions, methanogenic energy loss is still a very real energetic consequence that proximately derives from the microbiome. Similarly, if malnutrition leaves animals vulnerable to infection or dysbiosis in the gut, the microbiome still has a causal connection to host health. This then becomes a discussion about whether diet is an ultimate versus proximate mechanism, and the relative importance of other contributions to microbiome variation. (L536).
- **L530 “In relation to potential abrupt vs gradual changes in the microbiome preceding horse death” It seems unlike to make associations between changes in the gut microbiota and horse death considering that many samples were collected months or even years before the horse death. Similarly, some horses could have been sampled when the horse was suffering a disease process but which did not end up in the horse death. All these situations indicate the presence of many experimental bias which can limit the possibility of drawing robust conclusion.**

>> As noted earlier, we now include analyses demonstrating that the microbiome features most strongly associated with horse survival are also the features that are most repeatable within individuals across 7 years of sampling (L429). We further clarify on L108 that although samples were collected months preceding horse death, they were collected during a critical window of energy accumulation when we hypothesize that the microbiome is especially important.

We agree with the reviewer that the unknown of whether microbiome variation in the years prior to horse death can be predictive of mortality is an interesting question—this is precisely the prediction we sought to test in the analyses referenced by the reviewer. Indeed, we find that, to the extent that the microbiome does change prior to horse death, most features do so only in the year immediately preceding death (L437). Overall, this would seem to support the hypothesis proposed by the reviewer. However, there were also notable outliers from this pattern (L446), which speaks to the value of considering both acute and long-term perspectives of microbiome change.

The study of senescent patterns (distinct from aging) in wild animal populations represents a rather large field of study, and senescent patterns in the human microbiome have been well characterized. The likelihood that senescent patterns exist in the microbiome of horses therefore seems high, since immunosenescence across multiple years is one commonly observed pattern in long-lived mammalian wildlife (Froy et al. 2019, Science). Given that the gut microbiome is strongly regulated by host immunity, immunosenescence is one example of a mechanism that could cause detectable multi-year change in the microbiome preceding host death.

As noted in our response to a previous comment, scenarios like what the reviewer suggests (where horses were suffering from a disease process or had high abundance of mortality-associated microbiome features but did not die) would add noise to our

analyses, weakening rather than statistically biasing our ability to detect associations between the microbiome and horse survival. As we note above, the purpose of studying host-microbiome associations in a long-term individual-based study is to determine whether microbiome-fitness associations can be detected despite this real-world noise and complexity. The fact that we are able to quantify such associations, despite real-world noise and complexity, speaks to the strength of the host-microbiome relationship, and the impact of our research findings.

- **It should be identified the reason of each horse deaths in order to avoid potential bias. For example, a lack of gut microbial disbiosis would be expected if a horse die due to an accident.**

>> This is not possible for any long-term individual based ecological study, nor is it necessary for the research question we pursued. Despite the inability to track the precise cause of mortality for every study individual, decades of long-term studies have nonetheless been able to directly estimate the strength of natural selection acting on trait variation within wild populations. These studies provide the foundation for, and have disproportionately contributed to, our understanding the evolutionary biology of plant and animal populations (Clutton-Brock & Sheldon, Trends in Ecology and Evolution, 2010). This high bar set by the reviewer would exclude nearly all empirical studies of evolution by natural selection in the wild.

We further note that, while we cannot know the exact cause of death for every horse, we can be confident of every horse's death given this is a closed population which we exhaustively survey. This is a notable advantage when compared to the vast majority of long-term individual-based study systems, making Sable Island perhaps one of the best study systems in which to study relationships between the microbiome and host fitness.

Finally, we note that if 'accidents' unrelated to the microbiome are a primary cause of horse mortality, then this would have masked our ability to detect associations between the gut microbiome and horse survival (i.e., the microbiome would not be under selection)—this was not the case. Stated another way, mortality events that are unrelated to microbiome-linked starvation or nutritional deficits will have contributed noise to our dataset, rather than caused the patterns we observed. We are uncertain what 'accidents' could mean in the context of Sable Island, but note that, the rate of such events does not appear ecologically relevant, as indicated by the strength of the survival associations we were able to measure. Additionally, among periodic necropsy surveys conduct in early spring, we indeed find that horse mortality is predominately related to energy deficits.

- **L124 All herbivores prefer eating highly digestible forage instead of highly fibrous forage if they have the chance, as this imply a lower digestive effort. In ruminants a decrease in gut diversity is observed when animals consume highly digestible diets. Therefore, this may indicate that horses**

with lower diversity may eat the best grass (more digestible) and ultimately have a higher overwinter survival. On the contrary, the weak horses may have problems to reach the best grass (due to walking or group hierarchy limitations). This observation would explain why horses with a higher survival rate have a lower diversity and abundance of methanogens and ciliates (which are associated to de digestion of fibrous feeds), whereas horses with a lower survival rate seems to be more adapted to eat fibrous diet (possibly because they are unable to reach the best grass). If this is true, the differences in the microbiota must be considered as a consequence of the diet but not the reason to explain the horse survival.

>> As noted earlier, we now include analyses demonstrating that the microbiome features most strongly associated with horse survival are also the features that are most repeatable within individuals across 7 years of sampling (L429). Our results therefore do not support the interpretation that the survival-associations we observed are transient and primarily attributable to diet.

As noted earlier, we now provide analyses of dietary metabarcoding data, in which we find no evidence for a relationship between microbiome and diet dissimilarity among Sable Island horses (L526; Supplementary Materials; Supplementary Figure 5a,b). See earlier comment for a fuller discussion of this topic.

Decades of research has demonstrated a causal connection between methane emissions and energetic loss. It is extremely unlikely that variation in methane emissions between individuals are wholly attributable to diet variation. In fact, substantive variation in methane emissions exist among livestock fed identical diets. Methane-associated microbiota have also consistently emerged as among the most highly heritable microbiome features across human, livestock, and wild animal populations, and methane emissions are themselves heritable (L540). This provides the basis for animal breeding programs around the world which are now selecting for low methane emitting genotypes. Recent experimental transplants of bacteria and methanogens into gnotobiotic sheep fed identical diets further demonstrate that methanogenic outcomes strongly derive from priority effects and competitive interactions between gut microbiota (Yeoman et al. 2021, mBio). We now discuss this on L547.

The reviewer's concerns about horses being unable to reach quality forage, is not reflective of the ecology of this population, in which individuals of poor and good condition co-graze. Horses on Sable Island segregate into social groups/bands, and previous research has shown that group identity does not explain any variance in body condition scores after having taken age, sex, and longitude into account (Table 2 in Debeffe et al. 2016, Parasitology). Across decades of research on this population, we have yet to find evidence that weak horses are pushed away from the best grasses at a fine spatial scale. While we do observe some heterogeneity in plant communities and biomass across the Island (Supplementary Material), this environmental gradient can be

accounted for by longitude within our models (Supplementary Figure 5c,d). Furthermore, associations between the microbiome and horse survival appears similar across the island. We now also provide panoramic photographs from across the island (Fig. 1) to provide the reader with context as to the plant biomass available to the horses.

- **L132 A deviation of an individual's microbiome composition from the population's average does not necessarily imply a dysbiosis. It may indicate an adaptation to a specific environment or diet.**

>> We agree with the reviewer that deviation of microbiomes from the population centroid does not guarantee dysbiosis. As we now clarify on L150, we were testing a hypothesis proposed by Zaneveld et al. (2017), in which they speculate that, in some cases it can. Given that we have generated the largest shotgun metagenomic dataset derived from a wild population, we feel it is valuable to test major hypotheses pertaining to the host-microbiome relationship that pervade the literature, but for which (until now) we've lacked sufficient empirical data to evaluate. In this instance, analysis of our dataset would appear to support the reviewer's hypothesis in the context of gene family profiles, but not community composition, as we discuss on L154.

- **L137-140 It is well known that the diet is the main driver that determines de gut microbiota being much more relevant than other minor factors such as the geographical location, host genetics or even the animal species (a horse and a zebra eating the same diet tend to have a similar microbiota). Thus, these associations do not imply causality as they may be mediated by the type of diet consumed by the horse.**

>> As noted earlier, we now provide analyses of dietary metabarcoding data which do not support the prediction that the patterns we observed are caused by dietary variation.

Our findings are well-aligned with the vast majority of microbiome research which finds that, while diet can contribute to microbiome variation, its influence is often vastly over-estimated. The widespread mirroring of microbiome and host phylogenetic relationship (phylosymbiosis) is also a well characterized phenomenon. We therefore respectfully disagree with the reviewer's assertion that host genetics and differences between animal species are subordinate to diet effects. In recognition of the reviewer's concern that diet is the main driver of microbiome variation, we felt it could be beneficial to provide a brief cross-sectional overview of diet and genetic effects across human, lab animal, and wild animal systems, as well as directly test some of the points raised by the reviewer.

In a large study of 33 sympatric large-bodied mammalian herbivores in the East African savannah ecosystem, Kartzin et al. (2019, PNAS) found that microbiome variation is strongly correlated with host phylogenetic relatedness ($r = 0.91$) and only weakly correlated with diet relatedness ($r = 0.20$). Furthermore, dramatic dietary turnover in species between wet and dry seasons accounted for only $\frac{1}{4}$ of seasonal

turnover in the microbiome over the same period. Our own sampling took place within a very narrow seasonal context (little seasonal turnover in diet) and seasonal dietary variation would not have systematically biased our results, as horses which survived versus died were randomly sampled.

Knowles et al. (2019, Ecology Letters) likewise found that among mice, voles, and shrews sampled across disparate environments, host species family explained 35% of variation in the microbiome (Bray-Curtis dissimilarity) and trapping location upwards of 13%. In contrast, diet analysis based on gut contents found that diet similarity was not, or only extremely weakly, correlated with diet similarity among mice ($r = 0.07$, $p = 0.074$), shrews ($r = 0.10$, $p = 0.07$), and voles ($r = 0.22$, $p = 0.002$).

To the reviewer's own example of horses versus zebra, cursory analysis of publicly available equid microbiome data prepared and sequenced on the same sequencing runs (Edwards et al. 2020, Microbiome) reveals that—contrary to the reviewer's concern—clear differences in the microbiome exist between horses and captive zebra fed similar diets (Reviewer Response Figure 1; diet data available in paper supplementals). Samples from this study were sourced from horses and zebra spread across different locations, indicating that these species-effects are genuine, and not 'location' or 'technical' effects.

Reviewer Response Figure 1: A) An Aitchison distance principal component analysis ordination of 16S rRNA gene amplicon fecal microbiome data in domestic horses and zebra, and B) a plot of pairwise Aitchison distances across host-species contrasts.

In one of the largest human microbiome datasets assembled (samples from 8,208 individuals), Gacesa et al. (2022; Nature) find that, using 241 measurements of technical factors, anthropometrics, early-life and current exposome, self-reported diseases, use of medication, and diet, they are able to explain 12.9% of variation in the microbiome, and only 1% of total variation is attributable to diet. In contrast, additive genetics and co-housing each account for ~10% of variance in microbiome features, on

average (~20% cumulatively). Therefore, in this study, host additive genetics and social context (co-housing) had ~20x the effect of diet.

Diet does not appear to be the 'main driver' of microbiome variation among inbred lab mice kept under highly controlled laboratory conditions and exposed to extreme (non-ecologically realistic) diet manipulations, as one large meta-analysis of 27 diet-manipulation studies found that prevalent control versus high-fat/low-carbohydrate diet treatments accounted for between 3.5% to 11.6% of variation in microbiome dissimilarity, depending on the beta diversity measure considered (Bisanz et al., 2019, *Cell Host & Microbe*). Although notably, even these effects are likely over-estimations, since the authors only considered phylogeny-weighted beta diversity measures, and so likely underestimate heterogeneity and location-specific variation in ASV/OTU distribution.

Moeller et al. (2018; *Science*) demonstrate that clear location-specific signatures in the gut microbiome can be stably maintained across 11 generations of wild-derived mice bred in captivity and fed identical diets (highlighting the importance of host genetic effects and/or vertical microbial transmission).

Using embryo transplants of inbred mice into wild-derived mice, Rosshart et al. (2019; *Science*) experimentally demonstrate that lab-genotype mice born to wild mice acquire a 'wild' microbiome, and that the 'wild' microbiome acquired early in life is unaffected by diet manipulations (showing the importance of priority effects and early-life programming).

These examples are a small cross-section of the multitude of observational and experimental works which demonstrates that diet is often overemphasised as a determinant of microbiome structure. Host genetics can also be important (Ryu and Davenport, 2022, *Annual Review of Animal Biosciences*), and microbiome metacommunities and social transmission are increasingly being recognized as important drivers of microbiome variation (Raulo et al. 2020, *ISME*; Gacesa et al. 2022, *Nature*; Valles-Colomer et al. 2023, *Nature*; Sarkar et al. 2020, *Nature Ecology & Evolution*). Microbiomes are also now known to be highly idiosyncratic and repeatable within individuals through time (Björk et al. 2022, *Nature Ecology & Evolution*; Zhou et al. 2024, *Cell Host & Microbe*), partly due to genetics, but also priority effects and stabilizing interactions between microbiota themselves (Roche et al. 2023, *eLife*).

Our own current and previous work is consistent with other findings in the literature. For example, we (i) find spatial separation of individuals to be a key determinant of microbiome structure, (ii) do not find a strong effect of horse access to different plant forage species, and (iii) observe phylogenetic null-modelling evidence that microbial dispersal limitation (rather than heterogenous selective pressures due to diet) is the more likely mechanism underlying spatial turnover in the microbiome (Stothart et al. 2020, *Molecular Ecology*). Like Knowles et al. (2019, *Ecology Letters*) we do not find strong evidence for diet and microbiome dissimilarity in Sable Island horses

(Supplementary Materials; Supplementary Figure 5, L526). Like Björk et al. (2022, Nature Ecology & Evolution), we also find that components of the microbiome are repeatable within individuals across years (L429). Interestingly, our results also suggest that those features which are more consequential for horse survival also tend to be more repeatable (Figure 7a).

In summary, while diet can have statistically significant effects on the microbiome, we respectfully disagree that it is the 'main' driver of microbiome variation, or that it supersedes effects of 'geographical location, host genetics or even the animal species'. We hope that our brief overview of a broad cross-section of literature helps to clarify our perspective. We now include an abridged version of this discussion on L524.

- **L148 Can the AIC model be overfitted? It is well known that prediction models with a reasonable accuracy can be developed when using an internal validation process which implies the use of the same dataset to develop the model and to test its performance. On the contrary, an external validation, which implies the use of a large proportion of samples for model development (e.g. 90%) and the remaining samples for model validation, would imply a more robust determination of the model performance and it would prevent model overfitting (Fig S1). The inclusion of indicators of the model performance such as the determination coefficients of calibration R²C, cross validation R²CV, prediction R²P and their respective root mean square errors (RMSEC, RMSECV, RMSEP), the performance to prediction ratio (RPD) or the range in reference ratio (RER) would be appreciated.**

>> We now clarify that our purpose was not to develop a predictive tool, but rather, infer whether the gut microbiome is associated with horse survival in the wild by comparing competing hypotheses of microbiome form, function, or beta diversity dispersion against a null non-microbiome hypothesis (L159), and explore potential causal mechanisms which can be tested in future work (L176). The AIC approach we used penalizes each additional terms, and in doing so, helps to prevent over-fitting/parameterization. The cross-validation suggested by the reviewer would be most appropriate if prediction was our objective. For a more detailed overview of our reasoning and the philosophical differences between exploratory, inferential, and predictive studies, and for choosing the most appropriate statistical approach (including discussions of AIC and cross-validation), we refer to Tredennick et al. (2021), Ecology.

We are uncertain why Fig S1 is referenced as an example of model overfitting, as this was not a plot of actual versus predicted values. This was intended as a demonstration of how, using an even smaller subset of data (1 sample per individual) we observe the same general patterns of associations between the microbiome and horse survival.

- **L171-179 It is mentioned that food limitation is the most common cause of mortality in Sable Island horses. Thus, changes in the gut microbiota are most likely associated to insufficient feed intake or to the intake of low**

quality diet (fibrous diet) which is often associated to high abundance of methanogens and methanogenesis. Thus, changes in the gut microbiota cannot be considered as the cause of higher or lower survival as they are likely an indirect effect derived from other aspects such as physical, health or social horse limitations which can limit food intake or lead to changes in the diet (e.g. more fibrous diet). A chemical analysis of the horse faeces (e.g. fiber and protein content) and/or in vitro quantification of the CH₄ production after incubation of horse faeces with the same feed substrate across animals could help to proof this hypothesis. Alternatively, a quantification of the fermentation products (e.g. volatile fatty acids) from the horse faeces could provide useful information to link gut microbiota and function.

>> We now clarify that the food limitation we refer to occurs seasonally (i.e. little biomass during the winter) but is not limiting during the summer when horse microbiomes were sampled in the summer (L108). To help illustrate this, we now include panoramic photographs of the island landscape in Fig. 1, which are illustrative of the dense plant biomass available to horses over the summer. Sable Island horses are largely reliant on energy reserves amassed during the summer to survive the subsequent winter. We therefore predicted that it is this 'summer' microbiome that is critical for horse survival (L111), since inefficiencies or gut disruptions during this time would prevent individuals from accumulating the necessary fat/energy reserves. As previously mentioned in an earlier comment, we now include analysis of dietary metabarcoding data which do not support the prediction that variation in the Sable Island horse microbiome is primarily attributable to diet.

As suggested, we now provide analysis of short-chain fatty acid metabolomics data (L279, L381, L457). Analysis of these data strongly support the predictions made from our metagenomic dataset. We thank the reviewer for their suggestion to include these data.

As we are working with historically archived samples from a long-term study, it is not possible to retrospectively conduct the chemical and incubation experiments suggested. We agree this would be an exciting next step to further test our hypotheses in future years of data collection.

- **L198-208 When analysing metagenomics data, it is very common to find associations between gut microbes and genes as huge number of combinations are tested. However, those associations do not imply causality. This is particularly critical when assuming microbial effects on gut fermentation or CH₄ production without direct measurements of those parameters.**

>> We agree that performing a large number of tests can result in false positive findings, and this is why we applied a bias correction to account for multiple

comparisons (L183). Of course, this does not guarantee the elimination of all false positives and may even result in a greater number of false negatives. However, given the diversity and complexity of the microbiome, there is little viable alternative for an exploratory study of this nature. This approach is therefore extremely common across nearly all microbiome studies.

Because of the potential for false positive results, we deliberately refrained from focusing solely on associations between horse survival and any single microbiome feature. Instead, we opted to comprehensively consider the features that show the strongest associations with horse survival. In doing so, we find independent lines of evidence (different microbiota and different gene functions) that all suggest a connection between the microbiome the production of methane and horse survival. As suggested by reviewer 1, we now detail these 19 metagenomic associations in Supplementary Figure 4. We agree with the reviewer that these associations do not guarantee causality, however, we explain on L516, L522, L198 why a causal connection is likely. Publishing our results alongside our proposed mechanism lays the foundation for us, and other research groups, to begin more directly experimentally testing our hypothesis that methanogenesis may be a microbiome feature under selection among wild mammalian herbivores (L562).

- **Moreover, most of the concepts described here (H₂ sinks, sulphate or nitrate reduction) are derived from ruminant studies which have a different digestive physiology as they are foregut fermenters while horses are hindgut fermenters.**

>> The concepts we discuss and cite are derived from a broad cross-section of studies on ruminants, humans, hindgut fermenters, in vitro experimental studies, free-living environments, and lab mice. In fact, it is this recapitulation of the same metabolic patterns and methanogenic mechanisms across microbial communities that serves as the basis for our hypothesis.

We respectfully disagree with the reviewer that the metabolic pathways we discuss pertain only to ruminants. For example, (i) Hydrogenotrophic methanogens produce methane from H₂ and CO₂, whether they are in the rumen, intestines, or free-living environment, (ii) non-rumen sulfate and nitrate reducers will use H₂ while reducing sulfate/nitrate, and (iii) propionate/succinate is a hydrogen sink when compared to the most common pathways of acetate and butyrate synthesis etc. The metabolic pathways we discuss with respect to hydrogenotrophy are not specific to the rumen and have been well studied across a range of host, environmental, and artificial laboratory conditions. Similarly, the energetic cost of enteric methane emissions have been quantified across a wide array of mammalian herbivores, both wild and domestic (for example Clauss et al. 2020, Animal).

- **L212-226. Again, CH₄, propionate, lactate and fermentation products were not measured in this study. As a result, this paragraph is highly speculative.**

>> As suggested by the reviewer, we now provide analysis of short-chain fatty acid metabolomics data (L279, L381, L457, L465). Analysis of these data support the predictions made from our metagenomic dataset. We thank the reviewer for their suggestion to include these data.

- **L228 *Fibrobacter succinogenes* produces acetate and succinate as main fermentation products. As acetate production releases the highest amount of H₂ which is transformed into CH₄ by methanogens, it may imply an overall increase in CH₄ production. As indicated before, it has been observed that *F. succinogenes* is one of the most fibrolytic microbes in the gut and its abundance is link to the fibre intake. This again, highlights that the diet composition may be the key drier in the overwinter horse survival, while the effect in the gut microbiota may be indirect.**

>> As the reviewer suggests, *Fibrobacter succinogenes* produce acetate and succinate which we now clarify on L259 & L. However, succinate is known to be the major fermentation product, accounting for 70%-83% of the fermentation end-products produced by *F. succinogenes* (Gokarn et al. 1997, Applied Biochemistry and Biotechnology), and this pattern is consistent across growth substrates (Neumann, Weimer, and Suen 2018, Biotechnology for Biofuels). Furthermore, as now discussed on L290, the formation of acetate is thermodynamically unfavourable at high partial pressures of H₂, conditions where the risk of methanogenic energy loss are greatest.

Contrary to the reviewer's concerns that *F. succinogenes* support methanogenesis, despite being capable of acetate production, *F. succinogenes* have (i) been experimentally shown to competitively inhibit methanogenesis *in vitro* by competing for hydrogen (Asanuma et al. 1999, Journal of Dairy Science), (ii) been observationally linked to lower methane emissions among sheep (Stewart et al. 2019, Nature Biotechnology), and (iii) experimentally connected to lower methane emissions among gnotobiotic lambs (Chaucheyras-Durand et al. 2010, Applied and Environmental Microbiology). Metabolic models based on species-specific K_m estimates for dihydrogen further identify *F. succinogenes* as a key species capable of reducing H₂ flow to methanogenesis (Ungerfeld 2020, Frontiers in Microbiology). In short, *F. succinogenes* is a well-established model bacterial species for the competitive inhibition of methanogenesis in the mammalian gut.

As suggested by the reviewer, we now also provide analysis of short-chain fatty acid metabolomics data (L279, L381, L457, L465). Analysis of these data strongly support the predictions made from our metagenomic dataset. Namely, *Fibrobacterota* are negatively associated with butyrate concentrations, and associated with increases in

propionate concentrations (L279), as we predicted. We thank the reviewer for their suggestion to include these data.

Finally, we would kindly request clarification, since—assuming we have properly understood the reviewer—they appear to have raised contradictory concerns about the possible confounding effects of diet. In an earlier comment it was suggested that a high fibre diet will support greater methanogen abundance, thereby driving a negative association between horse survival and *Methanobrevibacter*—and that it is the high fibre content (rather than methanogens per se) that is proximately related to horse survival. If we understand properly, the reviewer is here suggesting that a high fibre diet is also positively related to *Fibrobacter succinogenes* abundance, and responsible for the positive association between horse survival and *Fibrobacter succinogenes* abundance. We are perhaps misunderstanding, in which case we apologize, but these scenarios seem to be mutually exclusive to us. Specifically, we are now unsure as to whether the reviewer is suggesting high fibre diets are positively or negatively associated with horse survival.

Nonetheless, as we now clarify on L533, if dietary fibre content was the proximate driver of horse survival, we would have expected gene contents related to fibre degradation (e.g., cellulase gene abundance) to be associated with survival (it was not). Similarly, we would have predicted fibre-degrading microbiota to show consistent, rather than divergent, associations with horse survival. As noted earlier, analysis of dietary metabarcoding data also do not support the interpretation that the patterns we observed are being chiefly driven by diet (L526).

- **L274-277 This sentence is highly speculative as fermentation products were not measured in this study.**

>> As suggested by the reviewer, we now also provide analysis of short-chain fatty acid metabolomics data (L279, L381, L457, L465). Analysis of these data strongly support the predictions made from our metagenomic dataset. We thank the reviewer for their suggestion to include these data.

- **Tables S4 to S7 are not provided.**

>> We thank the reviewer for raising concerns about this missing file. Tables S4 – S7 were uploaded in a separate appendix from the Supplementary Materials, since they contain a large volume of statistical outputs. We have ensured they have been uploaded with this resubmission. If they are again absent, we ask that the reviewer please contact the editor.

- **L342 The increase of protozoa during the year before death may indicate food scarcity which forced that horse to eat a high-fibrous diet as protozoa are fibrolytic microbes. This highlights that the type of diet consumed by the horse is linked to the survival rate, possibly as less fitted horses are pushed into the worst pastures with highly fibrous grasses.**

>> As suggested, we now clarify that the food limitation we refer to occurs seasonally (i.e. little biomass during the winter) but is not limiting during the summer when horse microbiomes were sampled (L108). To help illustrate this, we now include panoramic photographs in Figure 1, which are illustrative of the dense plant biomass available to horses over the summer. Sable Island horses are largely reliant on energy reserves amassed during the summer to survive the subsequent winter. We therefore predicted that it is this 'summer' microbiome that is critical for horse survival (L111), since inefficiencies or gut disruptions during this time would prevent individuals from accumulating the necessary fat/energy reserves. Therefore, insufficient food intake would not therefore explain the patterns we observed.

We now provide analyses of dietary metabarcoding data, in which we find no evidence for a relationship between microbiome and diet dissimilarity among Sable Island horses (L526; Supplementary Materials; Supplementary Figure 5a,b). Therefore, the patterns in the horse microbiome do not appear proximately attributable to differences in horse diet. Furthermore, what major dietary variation exists, can be largely accounted for by fitting longitude as a covariate within survival models (Supplementary Figure 5c,d). These results are consistent with our previous work, wherein we find using phylogenetic null models that dietary factors do not appear to be the main contributor to between-individual microbiome differences (Stothart et al. 2021; Molecular Ecology). To further test the robustness of our findings, we performed an additional set of selection analyses, after excluding horses from the western tip of Sable Island which have the most strongly differentiated diet profiles (attributable primarily to access to a low-fibre forage, sea sandwort; Supplementary Materials). Exclusion of these putative dietary outliers had superficial effects on the estimated associations between microbiome features and horse survival (Supplementary Figure 9).

We respectfully disagree with the reviewer's characterization of gut ciliates as fibrolytic specialists. As we understand the literature, ciliates are now known to be keystone predators within the gastrointestinal environment. For example, a large-scale meta-analysis found that the most repeatable effect of ciliate defaunation in the rumen environment across studies is to decrease ammonia concentrations, due to the removal of ciliate-based degradation of bacterial proteins (Newbold et al. 2015, *Frontiers in Microbiology*). In contrast, defaunation had nominal effects on fibre digestibility. Breakthroughs in techniques for cultivating rumen ciliates *in vitro* now reveal that differing competitive and predation pressures from ciliates shape prokaryotic community structure, metabolomic profiles, and methane emissions across cultures otherwise incubated on identical substrates (Solomon et al. 2022, *ISME*).

- **L361 Is should be defined what do you mean by “natural occurring variations in the hindgut microbiota”. For example, a lame or a nearly-blind or shy horse can have a different rumen gut microbiota in comparison with the population average, but this is driven by the diet as they are less fit and likely pushed to areas with the low-quality grass.**

>> As requested by the reviewer, we now clarify that ‘naturally occurring’ refers to microbiome variation within an unmanipulated free-living animal population in the wild (L132). This is in contrast to the vast majority of animal microbiome work, which is conducted in lab environments, domestic livestock, or human populations. The purpose of our work was to determine whether this variation was associated with horse survival outcomes, as predicted by hypotheses pertaining to the eco-evolutionary significance of host-associated microbiomes in nature. These hypotheses are widespread in the literature but have not been robustly empirically tested. This is the knowledge gap our work seeks to fill. See above for our use of dietary metabarcoding data which does not support the existence of a strong relationship between the gut microbiome and horse diet.

As detailed above and in our supplementary methods, the reviewer’s hypothesis of less fit horses being pushed to poorer pastures is not reflective of the ecology of this system, where large numbers of horses of varying condition are known to co-graze. What variation in ‘pasture quality’ or forage availability that exists on the island varies spatially from east to west and can be controlled for by fitting longitude as a covariate within our models. We further now demonstrate in the supplementary materials that diet composition is not strongly correlated with gut metagenomic variation.

- **L365 “causal dependence of survival on methanogenic pattern” can be a speculation as the CH₄ production was not measured and there are potential bias data interpretation.**

>> As suggested by the reviewer, we now clarify that we are proposing a hypothesis based upon observation of the microbiome features most strongly connected with horse survival (L186, L520, L562), and by drawing on decades of experimental evidence detailing the microbiome mechanisms underlying methanogenic energy loss. See earlier responses for descriptions of repeatability and dietary metabarcoding analyses which address the reviewer’s concerns about biased data interpretation.

- **L374-376 Most of the CH₄ production from livestock is coming from ruminants, therefore the horse, as a hindgut fermenter, cannot be used as an accurate model for ruminants.**

>> As suggested by the reviewer, we have removed this sentence as we agree it detracts from the main focus of our conclusions.

REVIEWERS' COMMENTS

Reviewer #1 (Remarks to the Author):

Thank you for addressing my comments.

Reviewer #3 (Remarks to the Author):

Authors have successfully answered all my questions and addressed my main concerns. I thank them for including additional information to clarify the main findings. Therefore, I do not have additional comments.